# Differentiation Through Black-Box Quadratic Programming Solvers

**Connor W. Magoon**[*1], **Fengyu Yang**[*1], **Noam Aigerman**[2], **Shahar Z. Kovalsky**[1]

[1] Department of Mathematics, University of North Carolina at Chapel Hill,
[2] Université de Montréal and Mila
[*] Equal contribution

## Abstract

Differentiable optimization has attracted significant research interest, particularly for quadratic programming (QP). Existing approaches for differentiating the solution of a QP with respect to its defining parameters often rely on specific integrated solvers. This integration limits their applicability, including their use in neural network architectures and bi-level optimization tasks, restricting users to a narrow selection of solver choices. To address this limitation, we introduce **dQP**, a modular and solver-agnostic framework for plug-and-play differentiation of virtually any QP solver. A key insight we leverage to achieve modularity is that, once the active set of inequality constraints is known, both the solution and its derivative can be expressed using simplified linear systems that share the same matrix. This formulation fully decouples the computation of the QP solution from its differentiation. Building on this result, we provide a minimal-overhead, open-source implementation (`https://github.com/cwmagoon/dQP`) that seamlessly integrates with over 15 state-of-the-art solvers. Comprehensive benchmark experiments demonstrate dQP's robustness and scalability, particularly highlighting its advantages in large-scale sparse problems.

## 1 Introduction

Computational methods often rely on solving optimization problems, *i.e.*, finding an optimum of an objective function subject to constraints. This has led to the development of numerous high-performance open-source and commercial solvers, particularly for constrained convex optimization [114, 27]. Recently, there has been growing interest in *differentiable optimization*, which enables gradient-based learning through optimization layers that show promise across applications such as image classification [8], optimal transport [97, 98], zero-sum games [75], tessellation [35], control [9, 38, 40], decision-making [110], robotics [62], biology [116], and NLP [111].

*Differentiable optimization* focuses on computing the derivatives of the optimal solution to an optimization problem with respect to parameters defining the problem's objective and constraints. Rooted in classical sensitivity analysis and parametric programming [99, 100, 47, 49, 48, 73, 26, 93], differentiable optimization has gained momentum through the idea that optimization problems encoding prior domain knowledge can be embedded as parameterized "layers" within neural networks – training such layers requires computing the derivative (gradient) of the optimization problem's solution during backpropagation [5, 1, 23].

This paper focuses on differentiating solutions to Quadratic Programs (QPs)—a core class of convex problems involving the minimization of a quadratic objective subject to linear inequality constraints.

Despite significant research efforts, existing approaches for differentiating QPs fail to fully leverage state-of-the-art solvers. General-purpose methods that support a broad class of optimization problems

39th Conference on Neural Information Processing Systems (NeurIPS 2025).

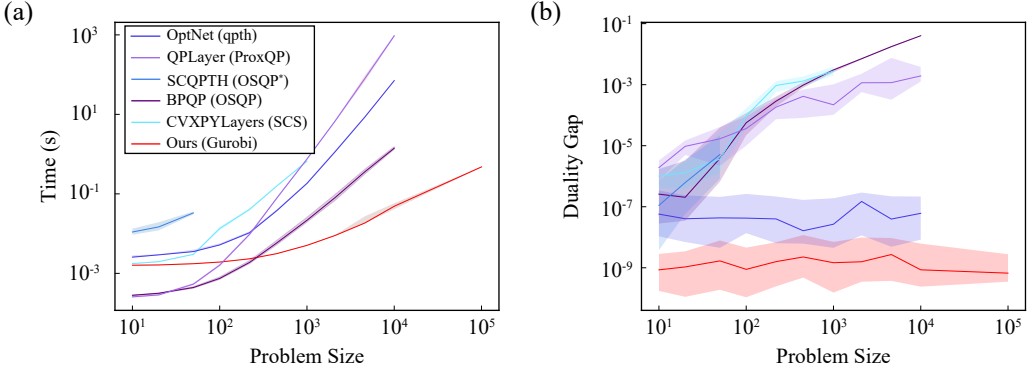

Figure 1: Comparison of differentiable QP methods for projection onto the probability simplex, evaluated by (a) total solve and differentiation time and (b) solution accuracy (duality gap). For moderately sized problems, our approach, using the Gurobi QP solver, outperforms existing methods on both metrics. As problem size increases, our method remains efficient, while others become intractable.

often include QPs, but they sacrifice efficiency and robustness for generality, underperforming compared to QP-specific methods. On the other hand, recent QP-specific approaches are tightly coupled with particular or proprietary solvers (*e.g.*, for batching small QPs), limiting their flexibility [5, 17, 29].

This tight integration restricts broader applicability. Solving QPs efficiently and reliably often requires advanced solvers like Gurobi [58] and MOSEK [10], developed over years of academic and commercial effort. These solvers handle scales and complexities that non-industrial implementations cannot. However, no single solver is optimal for all problems, making solver selection critical. To our knowledge, no existing approach provides a robust, efficient, and fully flexible framework and implementation for differentiating QPs across solvers.

We introduce **dQP**, a modular, solver-agnostic framework that transforms any QP solver into a differentiable layer. We strengthen modularity by introducing an explicit characterization of a QP's primal–dual solution in terms of its active constraint set. Our key insight is that once the active inequalities are known, both the solution and its derivative can be computed from simplified linear systems that share the same matrix. This formulation not only fully decouples optimization from differentiation but also enables differentiation of solvers that output only primal solutions, by completing the corresponding dual variables at negligible computational cost.

We implement dQP as an open-source, minimal-overhead layer on top of `qpsolvers` [32], enabling seamless integration with over 15 commercial and free solvers, and straightforward extension to additional ones. Our modular approach allows users to select the best solver for their task while providing differentiability. We evaluate dQP on a benchmark of over 2,000 diverse QPs and demonstrate superior performance compared to existing methods, particularly in large-scale structured sparse problems, as demonstrated in Figure 1.

**Our contributions:**

1. We design and implement a modular differentiable layer compatible with any QP solver. Our open-source code is available at `https://github.com/cwmagoon/dQP`.

2. We demonstrate state-of-the-art performance in solving and differentiating large-scale, sparse QPs.

3. We introduce an explicit, simplified characterization of a QP's solution and gradient in terms of its active constraints, which underpins our modular framework.

## 2 Related Works

**Differentiable Optimization.** OptNet [5, 4] differentiates QPs through their optimality conditions, and focuses on small dense problems for GPU batching. They solve the full Jacobian system efficiently by reusing the factorization employed in their custom interior-point method. However, as noted in

[17], this comes at the cost of ill-conditioning due to symmetrization. More recent differentiable QP methods include Alt-Diff and SCQPTH [109, 30, 29] which use first-order ADMM and approximately differentiate the fixed point map, and QPLayer [17] which accommodates infeasibility with extended conservative Jacobians. Similarly to OptNet, several of these are tightly integrated with custom algorithms, often to enable access to internal computations required for differentiation. Alt-Diff is coupled with a custom ADMM method, SCQPTH reimplements OSQP, and QPLayer is built on ProxQP. Several works have noticed the importance of the active constraint set in differentiating constrained optimization problems [8, 57, 89], as well as in the context of quadratic programming [9, 17, 86, 83]. A common observation is that the algebraic system obtained through implicit differentiation can be simplified by removing rows corresponding to inactive constraints. Some works have additionally observed that backpropagation itself can be cast as a distinct equality-constrained QP parameterized by the incoming gradients [8, 86]. However, existing approaches have not used these observations to create a differentiable layer that supports arbitrary black-box QP solvers, missing the opportunity to fully decouple optimization and differentiation. Other classes of optimization problems, such as convex cone programs [2] and mixed-integer programs [89], have also been differentiated. Some frameworks [1, 24, 92, 39, 96, 22, 90, 102] provide differentiable interfaces to broader classes of optimization problems, but their support of QP has significant limitations. CVXPYLayers [1] reformulates the QP into a cone program to use diffcp internally [2] and, as a result, does not support specialized QP solvers, relying exclusively on the cone solvers SCS, ECOS, and Clarabel. The framework Theseus [92] directly handles only unconstrained problems and similarly lacks support for QP-specific solvers. TorchOpt [96] and Optax [39] support unconstrained optimizers used in meta-learning, but require user-defined optimality conditions and user-supplied solvers to handle problems such as QPs. JAXopt [24] includes an implicit differentiation wrapper for CVXPY, which requires symbolic compilation of the QP and does not support sparse matrices; for sparse QPs, JAXopt includes a differentiable re-implementation of OSQP (similar to SCQPTH). Altogether, these drawbacks lead to subpar performance which is reported for some generic methods in previous work on differentiable QPs (*e.g.*, [17]). For completeness, Table 4 in Appendix A summarizes 24 relevant methods, including those discussed above.

**Implicit Layers.** Optimization layers are a class of implicit layers that use implicit differentiation to compute gradients of solution mappings that lack closed-form expressions [43]. Another class of implicit layer are deep equilibrium models, which are defined by fixed-point mappings and can be interpreted as infinitely deep networks [14, 66, 44, 59, 113, 15]. Similar techniques extend beyond algebraic equations to neural ODEs, where the adjoint state method from parametric PDE control is applied [76, 115, 18, 34]. Implicit differentiation also plays a key role in bi-level programming [36, 72, 56, 3] and meta-learning, where it enables optimization of the outer learning loop [50, 11, 61, 63, 95, 101]. Alternative methods bypass implicit differentiation using approximations – for example, by applying automatic differentiation to iterative algorithms via loop unrolling [19, 20, 81, 105], or by differentiating a single iteration or using Jacobian-free techniques for fixed-point mappings [53, 52, 25].

**Sensitivity Analysis and Parametric Programming.** There is extensive mathematical theory on the local behavior of optimization problems under perturbations [99, 100], particularly regarding solution sensitivity and stability [47, 49, 48, 73, 26]. While the implicit function theorem is central to this analysis, applying it to optimization problems requires an intermediate step: reformulating the problem through its optimality conditions, which demands several regularity assumptions. Sensitivity theory supports applications such as multi-parametric programming [93], including model predictive control, where problems must often be solved repeatedly for many parameter values, increasing computational costs. To address this, [21] observed that QPs admit closed-form solutions when the active set is known in advance, enabling offline precomputation. The parameter space can be partitioned into regions with fixed active sets [106], where the solution is stable under perturbations. This idea continues to inform modern methods [45, 82, 12].

# 3 Approach

We begin by formulating the problem and establishing the basic theory of QP differentiation. Then, we connect this foundation to our central theoretical observation and conclude with our straightforward algorithm derived from it. We note that various subparts of our discussion have been used to develop methods for differentiating QPs, see Section 2. However, no single work has fully developed the

explicit theory we present or leveraged it to practically implement a fully modular differentiable QP layer.

## 3.1 Problem Setup: Differentiating QPs

We consider a quadratic program in standard form,

$$
\begin{aligned}
z^*(\theta) = \arg\min_{z} \quad & \frac{1}{2} z^T P(\theta) z + q(\theta)^T z \\
\text{s.t.} \quad & A(\theta) z = b(\theta) \\
& C(\theta) z \leq d(\theta),
\end{aligned}
\tag{1}
$$

where $P \in \mathbb{R}^{n \times n}, q \in \mathbb{R}^n, A \in \mathbb{R}^{p \times n}, b \in \mathbb{R}^p, C \in \mathbb{R}^{m \times n}$ and $d \in \mathbb{R}^m$ are smoothly parameterized by some $\theta \in \mathbb{R}^s$. We assume that the QP is feasible and strictly convex (*i.e.*, $P \succ 0$). To simplify notation, in the following we omit $\theta$.

This work focuses on computing $\partial_\theta z^*(\theta) = \frac{\partial}{\partial \theta} z^*(\theta)$, the derivative of the optimal point of the QP (1) with respect to the parameters $\theta$. Intuitively, this derivative quantifies the change in the optimal point of the QP in response to a change in its parameters $\theta$. Our goal is to compute $\partial_\theta z^*(\theta)$ *efficiently* and *independently* of the method used to approximate the optimal point $z^*(\theta)$.

An important use case, highlighted in recent work on differentiable optimization, involves incorporating a QP of the form (1) as the $\ell$-th layer of a neural network. In this setting, the input to the QP layer, $x_\ell$, serves as the parameter vector $\theta$ that defines the QP's objective and constraints, and the output is the optimal solution $x_{\ell+1} = z^*(x_\ell)$. Generally, training such a network requires backpropagating gradients through each layer, which involves computing the Jacobian $\frac{\partial x_{\ell+1}}{\partial x_\ell}$. For a QP layer, this Jacobian corresponds to $\partial_\theta z^*(\theta)$, the derivative of the optimal point with respect to the problem parameters. This derivative is also crucial in descent-based methods for solving bi-level optimization problems [36].

## 3.2 KKT Conditions and Sensitivity Analysis

Our goal is to differentiate QPs using only the solution returned by a black-box numerical solver. To do so, we first establish the necessary theoretical foundations. As is standard in optimization, the required derivatives are obtained via sensitivity analysis of the KKT conditions. This section distills key ideas from optimization theory, sensitivity and parametric analysis, and differentiable programming, framing them in the context of QPs to support the development of dQP.

**Optimality Conditions.** The first-order Karush–Kuhn–Tucker (KKT) conditions [65, 71, 27, 114] provide a useful algebraic characterization of the optimal points of constrained optimization problems. For the QP (1), the KKT conditions take the form,

$$
\begin{aligned}
P z^* + q + A^T \lambda^* + C^T \mu^* &= 0 \\
A z^* - b = 0, \ C z^* - d \leq 0, \ \mu^* &\geq 0 \\
D(\mu^*)(C z^* - d) &= 0,
\end{aligned}
\tag{2}
$$

where $D(\mu^*) = \operatorname{diag}(\mu^*)$, and the additional variables $\lambda^* \in \mathbb{R}^p$ and $\mu^* \in \mathbb{R}^m$ are the optimal dual variables of the linear equalities and inequalities, respectively. The primal-dual solution of the QP (1) is defined by $\zeta^*(\theta) = (z^*(\theta), \lambda^*(\theta), \mu^*(\theta))$. Under strict convexity and feasibility, the QP (1) has a unique solution $\zeta^*(\theta)$, and the KKT conditions (2) are necessary and sufficient for its optimality.

**Active Set and Complementary Slackness.** The last equation in (2), the nonlinear complementary slackness condition, plays a central role in our work. Intuitively, it encodes the two possible states of each inequality constraint in (1), $(C z^* - d)_j \leq 0$. Either (i) the constraint is *active*, meaning it holds with equality $(C z^* - d)_j = 0$, in which case $\mu_j^* \geq 0$; or (ii) it is *inactive*, satisfied with strict inequality, in which case $\mu_j^* = 0$. Notably, if a constraint is inactive, the same optimal solution $z^*$ would be obtained even if that constraint were removed. We denote the active set by $J(\theta) = \{j : (C(\theta) z^*(\theta) - d(\theta))_j = 0\}$.

**Derivatives via Sensitivity Analysis.** To differentiate QPs, we apply the Basic Sensitivity Theorem (Theorem 2.1 in [46]), which underpins differentiation of the KKT conditions with respect to $\theta$.

Differentiability at $\theta$ requires the additional assumption of strict complementary slackness, ruling out the degenerate case where both $(Cz^* - d)_j = 0$ and $\mu_j^* = 0$, thus ensuring that the active set remains unchanged under small perturbations of $\theta$. Under this assumption, the primal-dual point $\zeta^*(\theta) = (z^*(\theta), \lambda^*(\theta), \mu^*(\theta))$ is differentiable in a neighborhood of $\theta$, optimal for the QP (1), uniquely satisfies the KKT conditions (2), and maintains strict complementary slackness. Crucially, the active set $J(\theta)$ remains fixed in this neighborhood.

With the active set stable, the equality conditions in (2) locally characterize $\zeta^{(}\theta)$. Implicit differentiation of these yields the Jacobians of the solution $\partial_\theta \zeta^*$ in terms of the linear system,

$$
\begin{bmatrix} P & A^T & C^T \\ A & 0 & 0 \\ D(\mu^*)C & 0 & D(Cz^* - d) \end{bmatrix} \begin{bmatrix} \partial_\theta z^* \\ \partial_\theta \lambda^* \\ \partial_\theta \mu^* \end{bmatrix} = - \begin{bmatrix} \partial_\theta P z^* + \partial_\theta q + \partial_\theta A^T \lambda^* + \partial_\theta C^T \mu^* \\ \partial_\theta A z^* - \partial_\theta b \\ D(\mu^*)(\partial_\theta C z^* - \partial_\theta d) \end{bmatrix}. \quad (3)
$$

Under the assumptions of the Basic Sensitivity Theorem, the linear system (3) is invertible. It degenerates exactly in the presence of weakly active constraints $\mu_j^* = (Cz^* - d)_j = 0$, for which the QP is non-differentiable (see, *e.g.*, [5]). For any inactive constraint $j \notin J$, the dual variable $\mu_j^*$ vanishes, and thus the corresponding rows and columns of (3) can be removed, yielding the simplified reduced form,

$$
\begin{bmatrix} P & A^T & C_J^T \\ A & 0 & 0 \\ C_J & 0 & 0 \end{bmatrix} \begin{bmatrix} \partial_\theta z^* \\ \partial_\theta \lambda^* \\ \partial_\theta \mu_J^* \end{bmatrix} = - \begin{bmatrix} \partial_\theta P z^* + \partial_\theta q + \partial_\theta A^T \lambda^* + \partial_\theta C_J^T \mu_J^* \\ \partial_\theta A z^* - \partial_\theta b \\ \partial_\theta C_J z^* - \partial_\theta d_J \end{bmatrix}, \quad (4)
$$

where $\mu_J^*$, $C_J$ and $d_J$ denote restriction to rows corresponding to active inequality constraints $j \in J$.

### 3.3 Extracting Derivatives from a QP Solver's Solution

With the above theory, we derive our main theoretical results and introduce **dQP**, a straightforward algorithm for efficient and robust differentiation of any black-box QP solver.

Our approach stems from two straightforward yet powerful insights: (1) given the *primal* solution of a QP, the active set can be easily identified; (2) once the active set is known, both the primal-dual optimal point and its derivatives can be derived explicitly in closed-form. Furthermore, these quantities can be computed efficiently via a single matrix factorization of a reduced-dimension symmetric system.

These observations lead to a simple algorithm: first, solve the optimization problem using *any* QP solver; then, identify the active set from the solution and solve a linear system to compute the derivatives. Consequently, we can define a "backward pass" for any layer that uses a QP solver, allowing for the seamless integration of any solver best suited to the problem, thus leveraging years of research and development invested in state-of-the-art QP solvers.

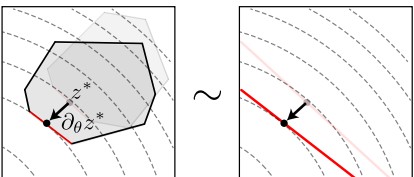

Figure 2: Illustration of active set differentiation. Left: a QP is shown by its quadratic level sets and polyhedral feasible set; the solution lies on a facet of the boundary; perturbations of the constraints lead to perturbations in the solution. Right: the perturbation of the solution remains the same when inactive constraints are eliminated.

**Locally Equivalent Equality-Constrained QP.** Consider the QP (1) and its optimal point $\zeta^*(\theta)$, along with the set $J(\theta)$ of active constraints, see Section 3.2. We define the reduced equality-constrained quadratic program, obtained by removing inactive inequalities and converting active inequality constraints into equality constraints,

$$
\begin{aligned}
z^*(\theta) = \arg\min_z \quad & \frac{1}{2} z^T P(\theta) z + q(\theta)^T z \\
\text{s.t.} \quad & \begin{bmatrix} A(\theta) \\ C(\theta)_{J(\theta)} \end{bmatrix} z = \begin{bmatrix} b(\theta) \\ d(\theta)_{J(\theta)} \end{bmatrix}.
\end{aligned} \quad (5)
$$

Under the assumptions of Section 3.2, this simpler QP is, in fact, *locally* equivalent to the QP (1), as illustrated in Figure 2. Moreover, it provides an explicit expression for both the primal-dual optimal point and its derivatives:

**Theorem 3.1.** *The QP* (5) *is locally equivalent to the reduced equality-constrained QP* (1) *and its solution* $\zeta^*(\theta) = (z^*(\theta), \lambda^*(\theta), \mu^*(\theta))$ *admits the explicit form*

$$\begin{bmatrix} z^* \\ \lambda^* \\ \mu_J^* \end{bmatrix} = \begin{bmatrix} P & A^T & C_J^T \\ A & 0 & 0 \\ C_J & 0 & 0 \end{bmatrix}^{-1} \begin{bmatrix} -q \\ b \\ d_J \end{bmatrix}. \tag{6}$$

*Furthermore, the optimal point can be explicitly differentiated to obtain*

$$\begin{bmatrix} \partial_\theta z^* \\ \partial_\theta \lambda^* \\ \partial_\theta \mu_J^* \end{bmatrix} = \begin{bmatrix} P & A^T & C_J^T \\ A & 0 & 0 \\ C_J & 0 & 0 \end{bmatrix}^{-1} \left( \begin{bmatrix} -\partial_\theta q \\ \partial_\theta b \\ \partial_\theta d_J \end{bmatrix} - \begin{bmatrix} \partial_\theta P & \partial_\theta A^T & \partial_\theta C_J^T \\ \partial_\theta A & 0 & 0 \\ \partial_\theta C_J & 0 & 0 \end{bmatrix} \begin{bmatrix} z^* \\ \lambda^* \\ \mu_J^* \end{bmatrix} \right). \tag{7}$$

A proof of this Theorem, based on the Basic Sensitivity Theorem [46], is provided in Appendix B, along with a calculation of the derivatives using differential matrix calculus [91, 78]. We note that this result is closely related to analyses studied in multi-parametric programming [21, 93, 106, 12, 82].

**Explicit Differentiation.** Notably, for quadratic programming, the Basic Sensitivity Theorem allows us to bypass implicit differentiation techniques [70]. We emphasize that this does not imply that the general solution or its active set admit a closed-form expression. Rather, while we perform explicit differentiation, the implicit function theorem remains key in establishing the local equivalence between the original and reduced problems -— the resulting derivatives are the same, though the derivations differ. The derivatives in (7) are obtained via ordinary (explicit) differentiation of the closed-form solution to the reduced QP (6), whereas those in (4) arise from implicit differentiation of the full nonlinear KKT conditions (2), followed by pruning inactive constraints. This distinction, formalized in Theorem 3.1, underscores a critical computational insight: once a black-box solver returns the primal solution, the active set can be identified and the derivatives computed directly via (7). Moreover, if the solver returns only the primal solution, the corresponding dual variables can be recovered using (6). Since both (6) and (7) rely on the same KKT matrix $K_J$, a single matrix factorization (e.g., via SuperLU [74]) suffices for both steps, adding negligible overhead. These insights culminate in the core algorithm of dQP, summarized in Algorithm 1.

**Numerical Computation.** Our approach enables compact and efficient gradient computation. The linear system in (7), used to compute both derivatives and dual variables, is symmetric and of reduced size. In contrast, implicit differentiation of the full KKT conditions (1) yields a significantly larger, asymmetric system (3). Beyond simplifying the derivative computation, our approach enables the use of fast, specialized linear solvers that exploit the reduced systems symmetric indefinite KKT matrix structure (*e.g.*, using an LDL factorization as in QDLDL [108, 37]).

Empirically, we observe that the reduced linear system (7) is often significantly better conditioned than its full counterpart. Figure 3 illustrates this with an example of a QP governed by two parameters $\theta = (\theta_1, \theta_2)$ of [106], calculated using DAQP [13]. Eliminating inactive constraints improves conditioning significantly.

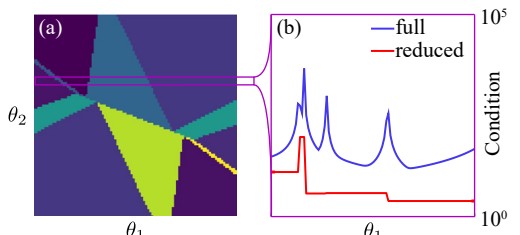

Figure 3 also highlights a key challenge: computing derivatives near non-differentiable singularities where the active set changes and some constraints become weakly active. In such regions, implicit differentiation becomes severely ill-conditioned. Our method is also affected, primarily through difficulty in determining the correct active set at an approximate solution. Several strategies have been proposed for improving active set identification [33, 84, 28]. Our implementation includes an optional refinement heuristic (Appendix D), though in all our experiments (Section 4), simple hard thresholding of the primal residual $r_j = (Cz^* - d)_j \geq -\epsilon_J$ proved robust.

Figure 3: (a) Active-set parameter space, coloring regions where the active set is constant. (b) Condition number of the full and reduced linear systems along a line in parameter space.

**Implementation.** Our open-source implementation is available at `https://github.com/cwmagoon/dQP`. We implement **dQP** (Algorithm 1) as a fully differentiable PyTorch module [88], providing an intuitive interface for integrating differentiable QPs into machine learning and bi-level

**Algorithm 1 – dQP**: Differentiation through Black-box Quadratic Programming Solvers

---

**Input:** $P, q, A, b, C, d$, and tolerance $\epsilon_J$
**Output:** $z^*, \lambda^*, \mu^*$ and $\partial_\theta z^*, \partial_\theta \lambda^*, \partial_\theta \mu^*$

1: Solve QP (1) with any solver for the primal solution $z^*$ (and $\lambda^*, \mu^*$ if available)
2: Compute the active set by thresholding: $\qquad\qquad\qquad J = \{j : (Cz^* - d)_j \geq -\epsilon_J\}$

3: Factorize the reduced KKT system matrix: $\qquad\qquad\qquad K_J = \begin{bmatrix} P & A^T & C_J^T \\ A & 0 & 0 \\ C_J & 0 & 0 \end{bmatrix}$

4: Compute $\lambda^*, \mu^*$ (if not obtained in step (1)): $\qquad\qquad \begin{bmatrix} z^* \\ \lambda^* \\ \mu_J^* \end{bmatrix} = K_J^{-1} \begin{bmatrix} -q \\ b \\ d_J \end{bmatrix}$

5: Compute the derivatives: $\quad \begin{bmatrix} \partial_\theta z^* \\ \partial_\theta \lambda^* \\ \partial_\theta \mu_J^* \end{bmatrix} = K_J^{-1} \left( \begin{bmatrix} -\partial_\theta q \\ \partial_\theta b \\ \partial_\theta d_J \end{bmatrix} - \begin{bmatrix} \partial_\theta P & \partial_\theta A^T & \partial_\theta C_J^T \\ \partial_\theta A & 0 & 0 \\ \partial_\theta C_J & 0 & 0 \end{bmatrix} \begin{bmatrix} z^* \\ \lambda^* \\ \mu_J^* \end{bmatrix} \right)$

---

Table 1: Performance of differentiable QP methods for 129 sparse problems from the Maros-Meszaros (MM) dataset [79].

| Solver | Full Dataset | | | | | | Subset of Problems Solved by All Methods | | | | | |
|---|---|---|---|---|---|---|---|---|---|---|---|---|
| | # Probs Solved | Avg Fwd [ms] | Avg Bwd [ms] | Avg Total [ms] | Avg Bwd/Total [%] | Accuracy [duality gap] | # Probs Solved | Avg Fwd [ms] | Avg Bwd [ms] | Avg Total [ms] | Avg Bwd/Total [%] | Accuracy [duality gap] |
| dQP (QPBenchmark) | **129** | 471 | 996 | **1467** | 57% | **$7.39 \times 10^{-6}$** | 24 | **10** | 83 | **93** | 35% | **$1.73 \times 10^{-7}$** |
| QPLayer (ProxQP) | 77 | 15089 | **632** | 15721 | 18% | $2.21 \times 10^{-2}$ | 24 | 2828 | 433 | 3261 | 29% | $1.77 \times 10^{-4}$ |
| OptNet (qpth) | 38 | 39329 | 2139 | 41468 | **6%** | $2.36 \times 10^{-3}$ | 24 | 9199 | 559 | 9758 | **7%** | $1.71 \times 10^{-4}$ |
| SCQPTH (OSQP*) | 55 | 16344 | 6551 | 22895 | 13% | $1.81 \times 10^{-2}$ | 24 | 14048 | 3019 | 17067 | 14% | $8.75 \times 10^{-3}$ |

optimization workflows. The implementation supports both dense and sparse problems end-to-end, with appropriate QP and linear solver support. As a PyTorch module, (7) must be rendered as a backward pass; this is described in Appendix C. To ensure modularity, the forward pass supports any QP solver interfaced via the open-source *qpsolvers* library [32], which provides lightweight access to over 15 commercial and free open-source solvers and easily supports the integration of new ones. We likewise offer flexibility in the choice of linear solver for differentiation, including support for large-scale sparse solvers like Pardiso [103] and symmetric indefinite solvers like QDLDL [108, 37]. For users unsure of which solver to use, dQP includes a profiling tool to help identify the best-performing QP solver for a given problem. Additional implementation details such as constraint normalization, handling non-differentiability, and warm-starting options for bi-level optimization are discussed in Appendix D.

## 4 Experimental Results

We have extensively tested dQP to ensure its robustness and evaluate its performance against existing methods for differentiable quadratic programming. Notably, we demonstrate dQP's strengths in handling large-scale structured and sparse problems, thus complementing custom differentiable GPU-batched solvers such as OptNet [5], which are optimized for solving many small, dense problems simultaneously. Given this focus, and considering the limited availability of state-of-the-art GPU-batchable QP solvers, we conduct our experiments on CPUs, similar to prior works [17, 29, 109, 1]. Our evaluation includes a large benchmark of over 2,000 challenging sparse and dense QPs taken from public and randomly generated problem datasets, designed to test dQPs robustness and performance. Additional experimental details are provided in Appendix E.

**Modularity and Performance.** We tested dQP on the QP Benchmark suite [31], focusing on 129 large-scale sparse problems from the standard Maros-Meszaros (MM) dataset [79], which are widely used as stress tests for QP solvers. We compared dQP against other differentiable QP methods available in the PyTorch framework: OptNet [5], QPLayer [17], SCQPTH [29], and CVXPYLayers [1]; using the authors' open-source implementations, each paired with its respective solver. We did not include other general frameworks that either lack direct support for generic QPs in PyTorch, showed subpar performance in our preliminary tests (consistent with findings reported in prior work [17]), or rely on forward passes built on diffcp/CVXPYLayers (see Table 4). For dQP's forward pass, we selected the top-performing QP solver for each problem, as identified by QP Benchmark [31], to demonstrate the importance of enabling differentiation for the solver that is best suited to each task.

The scatter plot in Figure 4 shows total runtime (forward + backward), duality gap (accuracy), and problem dimension (illustrated by point size) for each differentiable solver and problem. Aggregate

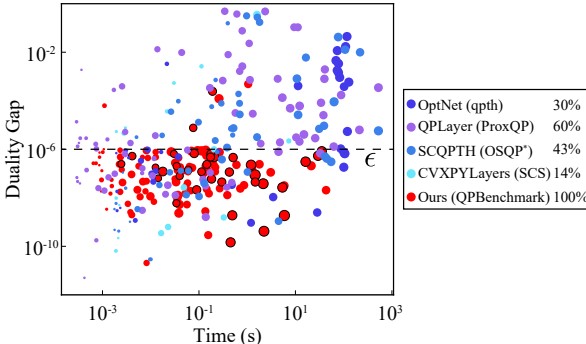

Figure 4: Accuracy versus total forward/backward solve for the Maros-Meszaros dataset [79]. Each point represents a solved problem; point size illustrates dimension; problems solved solely by dQP are circled. The legend shows percentages of success rates; the solvers dQP used and their counts are PIQP 100, Gurobi 9, ProxQP 9, Clarabel 7, OSQP 2, MOSEK 1, QPALM 1.

performance across the full dataset and the subset of problems solvable by all methods is summarized in Table 1. CVXPYLayers failed on all but 17 small-scale problems and is therefore excluded from the reported statistics. The MM dataset proved especially challenging, with OptNet and SCQPTH solving fewer than 50% of the problems. In contrast, dQP solved all MM problems and was the *only* method to succeed on 38 of them (circled in the figure). Moreover, dQP achieved the best total forward/backward runtime and accuracy in 80% and 78% of all problems, respectively. It performed particularly well on large-scale instances (dimension over 1000), being fastest and most accurate in 97% and 93% of such cases. Additional experiments on 450 random dense QPs and 625 sparse QPs (dimensions 10 to $10^4$) are provided in Appendix E.1.3, showing similarly strong performance.

**Scalability.** We evaluated dQP on large-scale sparse problems that are highly structured, a regime where state-of-the-art QP solvers have a significant advantage over less optimized solvers. Specifically, we tested dQP and other available differentiable QP solvers on two prototypical projection layers expressed as constrained QPs:

$$P_1(x) = \arg\min_z \|x - z\|_2^2 \quad \text{s.t.} \ 0 \leq z \leq 1, \sum z_i = 1, \tag{8}$$

and

$$P_2(x_1, \ldots, x_n) = \arg\min_{z_1, \ldots, z_n} \sum \|x_j - z_j\|_2^2 \quad \text{s.t.} \ \|z_j - z_{j+1}\|_\infty \leq 1. \tag{9}$$

Results for $P_1$ are shown in Figure 1, demonstrating dQP's scalability compared to OptNet and QPLayer. Other methods fail on all but small problems (see Appendix E.1.1). In dimensions greater than 2000, dQP outperforms competing methods by 2-3 orders of magnitude in both speed and accuracy. Competing methods are limited to dense calculations and fail in dimensions beyond $10^4$. We note that $P_1$ is the projection onto the probability simplex, also known as SparseMAX, for which more efficient, non-QP-based methods exist [80]. Results for $P_2$, representing projection onto "chains" with bounded links, exhibit similar scalability and are detailed in Appendix E.1.2.

Table 2: Performance analysis for projection onto the probability simplex, formulated in (8). Additional details are provided in Appendix E.1.1 and Table 5.

| Solver | Metric | Problem Size | | | | | | |
|---|---|---|---|---|---|---|---|---|
| | | 20 | 100 | 450 | 1000 | 4600 | 10000 | 100000 |
| dQP (Gurobi) | Accuracy | $\mathbf{1.07 \times 10^{-9}}$ | $\mathbf{8.88 \times 10^{-10}}$ | $\mathbf{2.26 \times 10^{-9}}$ | $\mathbf{1.47 \times 10^{-9}}$ | $\mathbf{2.72 \times 10^{-9}}$ | $\mathbf{9.55 \times 10^{-10}}$ | $\mathbf{6.67 \times 10^{-10}}$ |
| | Time [ms] | 1.63 | 1.92 | **3.13** | **5.06** | **18.46** | **49.00** | **476.64** |
| OptNet (qpth) | Accuracy | $4.04 \times 10^{-8}$ | $4.24 \times 10^{-8}$ | $1.64 \times 10^{-8}$ | $2.67 \times 10^{-8}$ | $3.95 \times 10^{-8}$ | $6.08 \times 10^{-8}$ | – |
| | Time [ms] | 2.92 | 5.19 | 37.66 | 182.99 | 8514.65 | 70856.43 | – |
| QPLayer (ProxQP) | Accuracy | $9.53 \times 10^{-6}$ | $3.65 \times 10^{-5}$ | $4.16 \times 10^{-4}$ | $2.19 \times 10^{-4}$ | $1.16 \times 10^{-3}$ | $1.94 \times 10^{-3}$ | – |
| | Time [ms] | **0.29** | 1.61 | 77.56 | 751.14 | 79314.49 | 946174.68 | – |
| BPQP (OSQP) | Accuracy | $2.04 \times 10^{-7}$ | $5.64 \times 10^{-5}$ | $9.66 \times 10^{-4}$ | $3.04 \times 10^{-3}$ | $1.76 \times 10^{-2}$ | $4.02 \times 10^{-2}$ | – |
| | Time [ms] | 0.32 | **0.75** | 5.81 | 21.68 | 358.78 | 1407.05 | – |
| CVXPYLayers (SCS) | Accuracy | $1.31 \times 10^{-6}$ | $9.47 \times 10^{-5}$ | $1.31 \times 10^{-3}$ | $2.79 \times 10^{-3}$ | – | – | – |
| | Time [ms] | 1.97 | 13.42 | 156.57 | 662.60 | – | – | – |
| SCQPTH (OSQP*) | Accuracy | $6.19 \times 10^{-7}$ | – | – | – | – | – | – |
| | Time [ms] | 14.75 | – | – | – | – | – | – |

Table 3: Performance analysis for projection onto chains, formulated in (9). Additional details are provided in Appendix E.1.2 and Table 6.

| Solver | Metric | Problem Size | | | | | | |
|---|---|---|---|---|---|---|---|---|
| | | 200 | 500 | 1000 | 2000 | 4000 | 10000 | 100000 |
| dQP (Gurobi) | Accuracy | $2.73 \times 10^{-7}$ | $2.02 \times 10^{-6}$ | $3.79 \times 10^{-6}$ | $9.16 \times 10^{-6}$ | $2.64 \times 10^{-5}$ | $4.29 \times 10^{-5}$ | $\mathbf{2.81 \times 10^{-4}}$ |
| | Time [ms] | **6.15** | **12.99** | **26.19** | **47.94** | **88.35** | **224.89** | **2432.64** |
| OptNet (qpth) | Accuracy | $\mathbf{6.97 \times 10^{-8}}$ | $\mathbf{1.75 \times 10^{-7}}$ | $\mathbf{9.22 \times 10^{-8}}$ | $\mathbf{2.43 \times 10^{-7}}$ | $\mathbf{2.60 \times 10^{-7}}$ | $\mathbf{1.98 \times 10^{-7}}$ | – |
| | Time [ms] | 25.38 | 169.64 | 907.25 | 5491.56 | 34799.98 | 571710.06 | – |
| QPLayer (ProxQP) | Accuracy | $8.46 \times 10^{-5}$ | $8.78 \times 10^{-5}$ | $1.82 \times 10^{-4}$ | $2.97 \times 10^{-4}$ | $6.95 \times 10^{-4}$ | $1.03 \times 10^{-3}$ | – |
| | Time [ms] | 8.04 | 81.93 | 577.11 | 3996.92 | 30748.67 | 471649.91 | – |
| SCQPTH (OSQP*) | Accuracy | $1.67 \times 10^{-5}$ | $2.83 \times 10^{-5}$ | $4.76 \times 10^{-5}$ | $6.64 \times 10^{-5}$ | $7.80 \times 10^{-5}$ | $1.21 \times 10^{-4}$ | – |
| | Time [ms] | 13.20 | 67.55 | 407.13 | 2755.28 | 16628.15 | 195462.18 | – |
| CVXPYLayers (SCS) | Accuracy | $1.69 \times 10^{-1}$ | $2.18 \times 10^{-1}$ | $2.97 \times 10^{-1}$ | – | – | – | – |
| | Time [ms] | 129.26 | 683.34 | 2048.34 | – | – | – | – |

**Bi-Level Geometry Optimization.** We further demonstrate the scalability of our approach on a bi-level optimization problem introduced in [69]:

$$M^* = \underset{M}{\arg\min} \ \|\mu^*(M)\|_2^2 \tag{10}$$
$$\text{s.t.} \ (v^*(M), \lambda^*(M), \mu^*(M)) = \underset{v}{\arg\min} \left\{ \text{tr}\left(v^T M v\right) \ \text{s.t.} \ Bv = u, \ CMv \succeq 0 \right\}.$$

In this problem $v \in \mathbb{R}^{n \times 2}$ represents the $n$ vertex coordinates of a triangular mesh, $M \in \mathbb{R}^{n \times n}$ is a parameterized Laplacian, and $B$, $u$ and $C$ encode boundary conditions. $\mu^*(M)$ is the dual variable corresponding to linear inequalities of the lower-level problem. The results described in [69] imply that if $\mu^*(M)$ vanishes then $v^*(M^*)$, at the optimal Laplacian $M^*$, represents a straight-edge intersection-free drawing of the mesh [112]. The solution to (10) is visualized in Figure 5(a) for an example large-scale ant mesh. Additionally, Figure 5(b) shows that dQP scales more favorably with problem size compared to OptNet, QPLayer and SCQPTH; in particular, only dQP supports problems with over $10^4$ vertices.

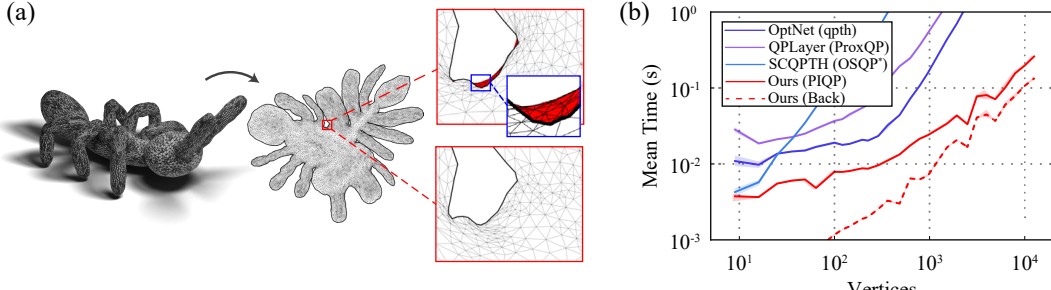

(a) (b)

Figure 5: (a) Planar embedding of a large-scale ant mesh (15k vertices). Zoom-ins show: (top) a non-injective harmonic map with edge overlaps highlighted in red; (bottom) dQP's injective solution to the bi-level problem (10). (b) Scalability: solver runtimes as mesh size increases for a synthetic problem.

## 5 Conclusion

We introduce the dQP framework for differentiating QPs by leveraging the local equivalence between a QP and a simpler equality-constrained problem. dQP provides a straightforward differentiable interface for any QP solver, yielding an efficient QP layer that can be readily integrated into neural architectures, among other applications. We note that our current method does not yet support full parallelization or GPU acceleration, as state-of-the-art sparse and scalable QP solvers with GPU support are still lacking. We recognize these as important challenges and limitations to address in future work. We see the present work as an important step toward developing similar solver-agnostic differentiable layers for other popular optimization problems (*e.g.*, semidefinite programming), which we plan to explore in future research.

## Acknowledgments

The authors gratefully acknowledge partial support from NSF grant DMS-2152289, FRG: Collaborative Research: Mathematical and Statistical Analysis of Compressible Data on Compressive Networks; the NSERC Discovery Grant RGPIN-2024-04605, Practical Neural Geometry Processing; and the FRQNT Établissement de la relève professorale grant 365040, Calcul rapide et léger des déformations à l'aide de réseaux neuronaux.

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

# A  Existing Methods for Differentiable Optimization

Table 4 provides an overview of existing differentiable optimization methods for quadratic programming (QP), more general conic programs (CP) and other optimization problems.

Table 4: Comparison of differentiable optimization libraries and layers.

| Name | Programs | Solvers | Features | Limitations |
|------|----------|---------|----------|-------------|
| OptNet [5] | QP | qpth | GPU batchable. Re-uses factorization from forward for fast backward solve. | Solver specific. Sparsity not supported. |
| QPLayer [17] | QP | ProxQP | Supports infeasible QPs. Tightly couples to ProxQP. | Solver specific. Sparsity not supported. |
| SCQPTH [29] | QP | OSQP | Differentiates ADMM updates. Supports infeasibility detection and automatic scaling. | Solver specific. Sparsity not supported. |
| Diffcp [2] | CP | SCS, ECOS, Clarabel | Supports sparse cone programs (CP). | Supports specific conic solvers. |
| CVXPYLayers [1] | CP | diffcp | Flexible problem formulation and solver choice. Supports sparsity. | Inherits diffcp limitations. |
| BPQP [86] | QP,SOCP | OSQP | Computes derivatives by solving a second equality-constrained QP. | Published implementation is incomplete. |
| Alt-Diff [109] | QP | Custom ADMM | Differentiates ADMM updates. Supports inexact solutions. | Reportedly slow [17]. |
| LQP [30] | QP | Custom ADMM | Differentiates ADMM updates. | Box constraints only. |
| LPGD [90] | CP | diffcp | Modifies diffcp to handle degenerate derivatives. | Inherits diffcp limitations. |
| CVXPYgen [102] | QP, SOCP | OSQP | Accelarates CVXPYlayers by C compilation of problem formulation. | Solver specific. |
| JAXopt-general [24] | User supplied | User supplied | Differentiates arbitrary implicit functions. | User must manually provide KKT.. |
| JAXopt-OSQP [24] | QP | Custom GPU OSQP | GPU batchable. | Solver specific. |
| JAXopt-CVXPYQP [24] | QP | CVXPY | Flexible problem formulation and solver choice. | Inherits CVXPY limitations. Sparsity not supported. |
| SDPRLayers [62] | CP | CVXPY | Flexible problem formulation and solver choice. | Inherits CVXPY limitations. |
| Torchopt [96] | User supplied | User supplied | Differentiates arbitrary implicit functions. | User must manually provide KKT. |
| Optax [39] | Specialized | Specialized | | Supports specialized problems/solvers, *e.g.*, projections. |
| DiffOpt.jl [22] | Various | JuMP-supported solvers [42] | Supports a variety of convex and non-convex problems. | JuMP provides QP support only for COSMO, OSQP, and Ipopt |
| Theseus [92] | NLS | CHOLMOD, cudaLU, BaSpaCho | Supports non-linear least squares (NLS). GPU batchable. Supports sparsity. | Lacks hard constraints. |
| OptNet-Sparse [7] | QP | qpth | | Published implementation is incomplete. |
| OptNet-CVXPY [6] | QP | CVXPY | | Published implementation is incomplete. |
| OSQPTh [107] | QP | OSQP | | Published implementation is incomplete. |
| DFWLayer [77] | QP | Frank-Wolfe | Automatically differentiates an unrolled Frank-Wolfe solver. | Solver specific. |
| CombOptNet [89] | ILP | Gurobi | Differentiation of combinatorial solvers | |
| Blackbox-Backprop [94] | MIP | Gurobi MIP, Blossom V, Dijkstra | Differentiation of combinatorial solvers | |

**Remarks.** This table includes all relevant differentiable optimization methods known to us at the time of writing. In some cases, code was available online without an associated publication. Among all listed methods that directly support generic QP without modification, only dQP, QPLayer, SDPRLayers, and DiffOpt.jl return both the dual optimal point of a QP and its derivatives. Notably, dQP is the sole QP method that supports a wide variety of solvers with minimal-overhead through *qpsolvers*, in contrast to a significant number of methods that build on top of CVXPY/CVXPYLayers, which requires a compilation step (*e.g.*, SDPRLayers, JAXopt-CVXPYQP, OptNet-CVXPY) and/or diffcp which has limited solvers compared to ordinary, non-differentiable CVXPY (*e.g.*, LPGD). Several methods are available only outside the PyTorch framework (*e.g.*, JAXopt, Optax, DiffOpt.jl).

# B Proof of Theorem 1

In this section we provide a proof of Theorem 1, which we restate below:

**Theorem B.1.** *The QP (5) is locally equivalent to the reduced equality-constrained QP (1) and its solution $\zeta^*(\theta) = (z^*(\theta), \lambda^*(\theta), \mu^*(\theta))$ admits the explicit form*

$$
\begin{bmatrix} z^* \\ \lambda^* \\ \mu_J^* \end{bmatrix} = \begin{bmatrix} P & A^T & C_J^T \\ A & 0 & 0 \\ C_J & 0 & 0 \end{bmatrix}^{-1} \begin{bmatrix} -q \\ b \\ d_J \end{bmatrix}.
\tag{11}
$$

*Furthermore, the optimal point can be explicitly differentiated to obtain*

$$
\begin{bmatrix} \partial_\theta z^* \\ \partial_\theta \lambda^* \\ \partial_\theta \mu_J^* \end{bmatrix} = -\begin{bmatrix} P & A^T & C_J^T \\ A & 0 & 0 \\ C_J & 0 & 0 \end{bmatrix}^{-1} \left( \begin{bmatrix} \partial_\theta P & \partial_\theta A^T & \partial_\theta C_J^T \\ \partial_\theta A & 0 & 0 \\ \partial_\theta C_J & 0 & 0 \end{bmatrix} \begin{bmatrix} z^* \\ \lambda^* \\ \mu_J^* \end{bmatrix} - \begin{bmatrix} -\partial_\theta q \\ \partial_\theta b \\ \partial_\theta d_J \end{bmatrix} \right).
\tag{12}
$$

*Proof.* We begin by establishing that the QP (1) and the equality-constrained reduced QP (5) are equivalent. For any $\theta$ satisfying the assumptions of the theorem, the QP (1) has a unique solution characterized by the KKT system

$$
\begin{aligned}
P(\theta)z^*(\theta) + q(\theta) + A(\theta)^T \lambda^*(\theta) + C(\theta)^T \mu^*(\theta) &= 0 \\
A(\theta)z^*(\theta) - b(\theta) &= 0 \\
C(\theta)z^*(\theta) - d(\theta) &\leq 0 \\
\mu^*(\theta) &\geq 0 \\
D(\mu^*(\theta))(C(\theta)z^*(\theta) - d(\theta)) &= 0.
\end{aligned}
\tag{13}
$$

Complementarity implies that active constraints $j \in J(\theta)$ have $\mu^*(\theta)_j > 0$ and therefore must be satisfied with an equality $(C(\theta)z^*(\theta) - d(\theta))_j = 0$, while inactive constraints $j \notin J(\theta)$ have $\mu^*(\theta)_j = 0$ and thus can be eliminated, without altering the solution. Therefore, the unique solution $\zeta^*(\theta) = (z^*(\theta), \lambda^*(\theta), \mu^*(\theta))$ of (13) is also the unique solution of the reduced system

$$
\begin{aligned}
P(\theta)z^*(\theta) + q(\theta) + A(\theta)^T \lambda^*(\theta) + C(\theta)_{J(\theta)}^T \mu^*(\theta)_{J(\theta)} &= 0 \\
A(\theta)z^*(\theta) - b(\theta) &= 0 \\
C(\theta)_{J(\theta)} z^*(\theta) - d(\theta)_{J(\theta)} &= 0,
\end{aligned}
\tag{14}
$$

which are exactly the KKT conditions of the equality-constrained reduced QP (5). Uniqueness of solution then implies that (1) and (5) are pointwise equivalent at $\theta$. Beyond this, since $P, q, A, b, C, d$ are smoothly parameterized by $\theta$, the Basic Sensitivity Theorem [46] asserts that the primal-dual solution $\zeta^*(\theta)$ for (1) is a differentiable function of $\theta$ in a neighborhood of $\theta$, defined implicitly through the KKT's equality conditions, and that the active set $J(\theta)$ is fixed in this neighborhood. Thus, (1) and (5) are locally equivalent in a neighborhood of the parameter $\theta$.

(14) implies that the reduced primal-dual solution $\zeta_J^*(\theta) = (z^*(\theta), \lambda^*(\theta), \mu_J^*(\theta))$ satisfies $K_J(\theta)\zeta_J^*(\theta) = v_J(\theta)$, where

$$
K_J(\theta) = \begin{bmatrix} P(\theta) & A(\theta)^T & C(\theta)_{J(\theta)}^T \\ A(\theta) & 0 & 0 \\ C(\theta)_{J(\theta)} & 0 & 0 \end{bmatrix}, \quad v_J(\theta) = \begin{bmatrix} -q(\theta) \\ b(\theta) \\ d(\theta)_{J(\theta)} \end{bmatrix}.
\tag{15}
$$

Under the assumptions of the theorem, the reduced KKT matrix $K_J(\theta)$ is invertible and

$$
\zeta_J^* = K_J^{-1} v_J,
\tag{16}
$$

yielding (6). Moreover, since $J(\theta)$ is constant in a local neighborhood, the Basic Sensitivity Theorem establishes that $\zeta_J^*(\theta)$ is differentiable. Using the derivative of the matrix inverse [78, 91], we *explicitly* differentiate (16) to obtain

$$
\partial_\theta \zeta_J^* = (-K_J^{-1}(\partial_\theta K)K_J^{-1})v_J + K_J^{-1}(\partial_\theta v_J) = -K_J^{-1}(\partial_\theta K_J)\zeta_J^* + K_J^{-1}(\partial_\theta v_J),
\tag{17}
$$

yielding (7).

$\square$

## C   Backpropagation

Like other differentiable QP layers implemented within automatic differentiation frameworks such as PyTorch [88], we do not directly return the derivative $\partial_\theta \zeta^*$. This is in part because dQP directly receives the QP parameters $P, q, A, b, C, d$ and not $\theta$, and so in backpropogation we are not concerned with $\theta$. This is rather accounted for in the next step outside dQP, usually by automatic differentiation. Secondly, backpropagation requires that we compute a so-called Jacobian-vector product which are products of the Jacobians with an "incoming" gradient of a quantity or loss $\ell$ that depends on $\zeta^*$. This requires less computation and does not require the formation of a 3-tensor. Since $\zeta_J^* = K_J^{-1} v_J$ is a formal matrix-vector multiplication, the Jacobian-vector product is well-known,

$$\nabla_{v_J} \ell = (K_J^{-1})^T \; \nabla_{\zeta_J^*} \ell, \tag{18}$$

and

$$\nabla_{K_J} \ell = -\nabla_{v_J} \ell \; \zeta_J^{*T}, \tag{19}$$

with respect to $K_J, v_J$, respectively. Although backpropagation introduces a transposition, the re-use of a factorization from solving for the active duals is unaffected. This follows from the symmetry of the reduced KKT matrix which simplifies (18) into $\nabla_{v_J} \ell = K_J^{-1} \nabla_{\zeta_J^*} \ell$. Next, we extract the gradients with respect to the parameters by the chain rule. This amounts to tracking their position in the blocks and accounting for symmetry constraints. It is helpful to write $(d_z, d_\lambda, d_{\mu_J}) = -\nabla_{v_J} \ell$ so that we express

$$
\begin{aligned}
\nabla_P \ell &= \frac{1}{2} \left( d_z z^{*T} + z^* d_z^T \right) & \nabla_q \ell &= d_z \\
\nabla_A \ell &= d_\lambda z^{*T} + \lambda^* d_z^T & \nabla_b \ell &= -d_\lambda \\
\nabla_{C_J} \ell &= d_{\mu_J} z^{*T} + \mu_J^* d_z^T & \nabla_{d_J} \ell &= -d_{\mu_J},
\end{aligned}
\tag{20}
$$

similar to OptNet [5]. We note that the gradient with respect to $P$ is constrained to lie within the subspace of symmetric matrices. Similarly, if the matrices $P, A, C$ are sparse, then we project the gradient to lie within the non-zero entries, which can be implemented efficiently in Equations 20. Although the above argument is for a scalar loss $\ell$, the same approach is naturally adapted if $\zeta^*$ is mapped to a vector in the immediate next layer.

## D   Implementation Details

**Choosing a solver.** Since our work enables users to choose any QP solver as the front-end for their differentiable QP applications, we include a simple diagnostic tool for quantitatively measuring solver performance. We present an example result in Figure 6 for the cross geometry experiment in Figure 9, finding PIQP, OSQP, and QPALM to be the most efficient. For this reason, we choose PIQP in the geometry experiments. We also include tools for checking the solution and gradient accuracy.

**Tolerances.** In addition to the active set tolerance $\epsilon_J$, QP solvers often support additional user-provided tolerances. These include the primal residual which measures violations of feasibility, the dual residual which measures violations of stationary, and for some solvers also the duality gap, which provides a direct handle on solution accuracy. We inherit the structure of *qpsolvers* for setting custom tolerances on different QP solvers, though we set a heuristic default which is sufficient for many of the experiments in this work.

**Convexity and Feasibility.** Two key assumptions of our method are strict convexity and feasibility. However, these are often violated in practice. We include optional checks that $P$ is symmetric positive definite. On the other hand, we do not perform any special handling for infeasibility – a limitation of our method compared to, for example, QPLayer [17].

**Non-differentiable Points.** For non-differentiable problems, we solve for the derivatives in the least-squares sense, plugging the system into *qpsolvers* which can handle least-squares, or a standard least-squares solver. We attempt to anticipate weakly active constraints which cause non-differentiability by measuring the norms of the primal residual and the dual. Additionally, the reduced KKT is non-invertible if the active dual solution is not unique and so we check a necessary condition: the total number of active constraints plus the number of equalities must be less than the dimension. If these checks are passed, we attempt the standard linear solve and pass to least-squares if it fails.

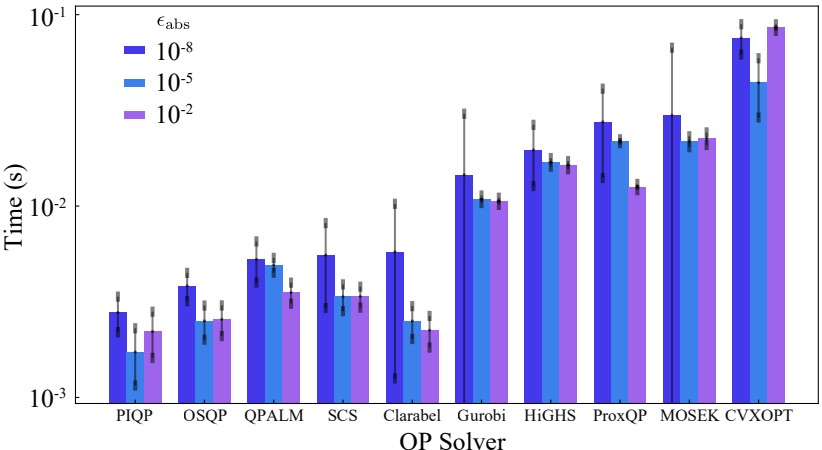

Figure 6: Evaluating the best QP solver for the cross geometry problem (Figure 9) using our diagnostic tool. The solution tolerance regimes are varied between $\epsilon_{abs} = 10^{-8}, 10^{-5}, 10^{-2}$.

**Normalization.** Some problems have large variations in scale between different rows within the constraints. This influences the primal residual and thus the active set, which is determined by comparing with an absolute threshold tolerance. To address this issue for these problems, we include an *optional* differentiable normalization step on the constraints before Algorithm 1 is carried out. Under this choice, the resulting relative primal residual becomes the scale-invariant distance to the constraint.

**Equality Constraints.** While we include equality constraints in our general formulation, they are not required.

**Warm-Start.** Since *qpsolvers* supports warm-starting, we inherit it as an option and store data in the PyTorch module from previous outer iterations, which can be used as initialization. This is useful for bi-level optimization problems where the input $\theta$ changes little between outer iterations. We did not use this feature in our bi-level optimization experiment.

**Fixed Parameters.** For fixed parameters, we do not compute the corresponding derivative. This saves the cost of unwrapping the linear solve as in (20) and saves the memory to form the loss gradients, which are matrices for $P, A, C$.

**Active Set Refinement.** Inaccuracy in a solution may lead to instability in the active set near weakly active constraints, degrading the gradient quality. To show this, we repeat the experiment in Figure 3 which has a simple polyhedral active set parameter space. One setup where instability appears is illustrated in Figure 7 where we use absolute solver tolerance $\epsilon_{abs} = 10^{-4}$ and a much tighter active tolerance $\epsilon_J = 10^{-7}$. Qualitatively, the active set at each solution is severely degraded, even for points away from the boundaries where the set changes. We provide a *optional* heuristic algorithm

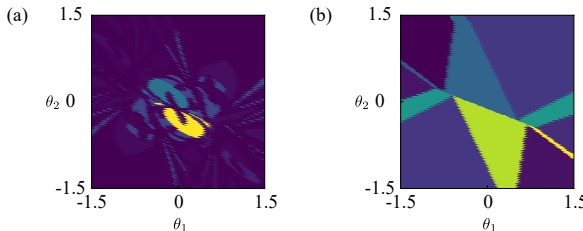

Figure 7: The set-up in figure 3 with looser solver tolerance $\epsilon_{abs} = 10^{-4}$, active tolerance $\epsilon_J = 10^{-7}$, and solver PIQP. (a) The computed active set is degraded due to the inaccurate solution. (b) Our heuristic active set refinement algorithm recovers the ground truth active sets.

to address this, which recovers the desired set in this problem. First, we order the constraints by increasing residual and select an initial active set from the tolerance $\epsilon_J$. Then, we progressively add

constraints by checking if the residual of the system 6 for $\zeta_J^*$ decreases, and greedily accepting until adding constraints no longer improves the residual. At each step, we keep the primal solution from the forward fixed, and solve for the new active dual variables. While this algorithm works well on simple examples, more sophisticated and efficient techniques may be desired for harder problems. We did not use this refinement algorithm in any of our experiments.

**QP Solvers.** Throughout this work, we use a number of QP solvers available in *qpsolvers* including Clarabel [55], DAQP [13], Gurobi [58], HiGHS [64], HPIPM [51], MOSEK [10], OSQP [108], PIQP [104], ProxQP [16], QPALM [60], qpSWIFT [87], quadprog [54], and SCS [85].

## E  Experimental Details

For completeness and reproducibility, we include additional details on the experiments. We run all experiments and methods on CPU, including methods that support GPU such as OptNet.

### E.1  Performance Evaluation

All experiments in this section were run on a Macbook Air with Apple M2 chips, 8 cores, and 16GB RAM.

In our QP benchmark experiments, we evaluate the solution accuracy using the primal residual $r_p$ (the maximum error on equality and inequality constraints), dual residual $r_d$ (the maximum error on the dual feasibility condition), and duality gap $r_g$ (the difference between primal and dual optimal values).

$$
r_p = \max \left( \|Az - b\|_\infty , [Cz - d]_+ \right)
$$
$$
r_d = \left\| Pz + q + A^T \lambda + C^T \mu \right\|_\infty
$$
$$
r_g = |z^T Pz + q^T z + b^T \lambda + d^T \mu|
$$

Throughout our experiments, we present results for the duality gap to indicate the solution accuracy since, for a strongly convex QP, a zero duality gap $r_g = 0$ is a necessary and sufficient condition for optimality.

For the forward, we set the absolute residual tolerance to $\epsilon_{\text{abs}} = 10^{-6}$ and the active constraint tolerance to $\epsilon_J = 10^{-5}$. We run each problem separately with batch size 1.

In our benchmark, we regard a problem as successfully solved if it meets the following criteria:

1. The solve time is less than a practical 800s time limit.

2. The primal residual, dual residual, and duality gap are less than 1.0. This is a coarse check, less stringent than the imposed tolerances.

3. The differentiation is executed, and does not lead to a fatal error (e.g. due to non-invertibility of a linear system).

Experimental results are averaged over 5 independent samples.

Since SCQPTH does not support equality constraints, we convert them into a corresponding set of inequality constraints.

### E.1.1  Projection onto the probability simplex

For projection onto the probability simplex, as formulated in $P_1$, equation (8), we set $x \in \mathbb{R}^n$ with $x_i \sim \mathcal{N}(0, 1)$ drawn randomly from a standard normal distribution. The dataset, with 500 problems, has dimensions $n \in \{10, 20, 50, 100, 220, 450, 1000, 2100, 4600, 10000, 100000\}$. For $n \leq 4600$, each dimension contains 50 problems and 25 problems for $n > 4600$. Gurobi is chosen as dQP's forward solver. Figure 1 shows the median performance within the $1/4$ and $3/4$ quantiles for each dimension. SCQPTH failed for all problems with $n > 50$. The statistics in Table 5 show that dQP outperforms competing methods for differentiable QP in both forward and backward times.

Table 5: Time and accuracy performance statistics for projection onto the probability simplex.

| Solver | Metric | Problem Size | | | | | | |
|---|---|---|---|---|---|---|---|---|
| | | 20 | 100 | 450 | 1000 | 4600 | 10000 | 100000 |
| dQP (Gurobi) | Accuracy | $\mathbf{1.07 \times 10^{-9}}$ | $8.88 \times 10^{-10}$ | $\mathbf{2.26 \times 10^{-9}}$ | $\mathbf{1.47 \times 10^{-9}}$ | $\mathbf{2.72 \times 10^{-9}}$ | $9.55 \times 10^{-10}$ | $\mathbf{6.67 \times 10^{-10}}$ |
| | Forward [ms] | 1.38 | 1.65 | **2.66** | **4.37** | **15.83** | **42.21** | **423.91** |
| | Backward [ms] | 0.24 | **0.28** | **0.46** | **0.69** | **2.58** | **6.21** | **53.45** |
| | Total [ms] | 1.63 | 1.92 | **3.13** | **5.06** | **18.46** | **49.00** | **476.64** |
| OptNet (qpth) | Accuracy | $4.04 \times 10^{-8}$ | $4.24 \times 10^{-8}$ | $1.64 \times 10^{-8}$ | $2.67 \times 10^{-8}$ | $3.95 \times 10^{-8}$ | $6.08 \times 10^{-8}$ | Failed |
| | Forward [ms] | 2.72 | 4.72 | 33.46 | 165.50 | 7788.73 | 65976.45 | – |
| | Backward [ms] | 0.20 | 0.46 | 3.99 | 17.48 | 720.43 | 4958.74 | – |
| | Total [ms] | 2.92 | 5.19 | 37.66 | 182.99 | 8514.65 | 70856.43 | – |
| QPLayer (ProxQP) | Accuracy | $9.53 \times 10^{-6}$ | $3.65 \times 10^{-5}$ | $4.16 \times 10^{-4}$ | $2.19 \times 10^{-4}$ | $1.16 \times 10^{-3}$ | $1.94 \times 10^{-3}$ | Failed |
| | Forward [ms] | 0.14 | 1.23 | 66.73 | 657.88 | 71724.25 | 869532.53 | – |
| | Backward [ms] | **0.14** | 0.37 | 10.85 | 91.72 | 7594.93 | 77831.58 | – |
| | Total [ms] | **0.29** | 1.61 | 77.56 | 751.14 | 79314.49 | 946174.68 | – |
| BPQP (OSQP) | Accuracy | $2.04 \times 10^{-7}$ | $5.64 \times 10^{-5}$ | $9.66 \times 10^{-4}$ | $3.04 \times 10^{-3}$ | $1.76 \times 10^{-2}$ | $4.02 \times 10^{-2}$ | Failed |
| | Forward [ms] | **0.11** | **0.40** | 3.88 | 14.45 | 226.27 | 712.20 | – |
| | Backward [ms] | 0.21 | 0.34 | 1.90 | 7.10 | 132.73 | 692.37 | – |
| | Total [ms] | 0.32 | **0.75** | 5.81 | 21.68 | 358.78 | 1407.05 | – |
| CVXPYLayers (SCS) | Accuracy | $1.31 \times 10^{-6}$ | $9.47 \times 10^{-5}$ | $1.31 \times 10^{-3}$ | $2.79 \times 10^{-3}$ | Failed | Failed | Failed |
| | Forward [ms] | 1.30 | 10.51 | 131.60 | 537.26 | – | – | – |
| | Backward [ms] | 0.66 | 2.92 | 23.83 | 123.25 | – | – | – |
| | Total [ms] | 1.97 | 13.42 | 156.57 | 662.60 | – | – | – |
| SCQPTH (OSQP*) | Accuracy | $6.19 \times 10^{-7}$ | Failed | Failed | Failed | Failed | Failed | Failed |
| | Forward [ms] | 14.35 | – | – | – | – | – | – |
| | Backward [ms] | 0.40 | – | – | – | – | – | – |
| | Total [ms] | 14.75 | – | – | – | – | – | – |

Table 6: Time and accuracy performance statistics for projection onto chains.

| Solver | Metric | Problem Size | | | | | | |
|---|---|---|---|---|---|---|---|---|
| | | 200 | 500 | 1000 | 2000 | 4000 | 10000 | 100000 |
| dQP (Gurobi) | Accuracy | $2.73 \times 10^{-7}$ | $2.02 \times 10^{-6}$ | $3.79 \times 10^{-6}$ | $9.16 \times 10^{-6}$ | $2.64 \times 10^{-5}$ | $4.29 \times 10^{-5}$ | $\mathbf{2.81 \times 10^{-4}}$ |
| | Forward [ms] | **5.66** | **12.04** | **24.41** | **44.79** | **82.57** | **209.79** | **2263.54** |
| | Backward [ms] | **0.49** | **0.98** | **1.74** | **3.18** | **5.81** | **14.69** | **172.80** |
| | Total [ms] | **6.15** | **12.99** | **26.19** | **47.94** | **88.35** | **224.89** | **2432.64** |
| OptNet (qpth) | Accuracy | $\mathbf{6.97 \times 10^{-8}}$ | $\mathbf{1.75 \times 10^{-7}}$ | $\mathbf{9.22 \times 10^{-8}}$ | $\mathbf{2.43 \times 10^{-7}}$ | $\mathbf{2.60 \times 10^{-7}}$ | $\mathbf{1.98 \times 10^{-7}}$ | Failed |
| | Forward [ms] | 23.37 | 156.49 | 845.24 | 5124.87 | 32528.54 | 536702.00 | – |
| | Backward [ms] | 1.98 | 13.06 | 61.41 | 365.02 | 2266.20 | 35438.33 | – |
| | Total [ms] | 25.38 | 169.64 | 907.25 | 5491.56 | 34799.98 | 571710.06 | – |
| QPLayer (ProxQP) | Accuracy | $8.46 \times 10^{-5}$ | $8.78 \times 10^{-5}$ | $1.82 \times 10^{-4}$ | $2.97 \times 10^{-4}$ | $6.95 \times 10^{-4}$ | $1.03 \times 10^{-3}$ | Failed |
| | Forward [ms] | 6.60 | 69.90 | 505.05 | 3484.47 | 26921.57 | 414295.22 | – |
| | Backward [ms] | 1.44 | 12.10 | 72.13 | 512.95 | 3833.25 | 57219.68 | – |
| | Total [ms] | 8.04 | 81.93 | 577.11 | 3996.92 | 30748.67 | 471649.91 | – |
| SCQPTH (OSQP*) | Accuracy | $1.67 \times 10^{-5}$ | $2.83 \times 10^{-5}$ | $4.76 \times 10^{-5}$ | $6.64 \times 10^{-5}$ | $7.80 \times 10^{-5}$ | $1.21 \times 10^{-4}$ | Failed |
| | Forward [ms] | 10.02 | 39.49 | 236.61 | 1617.88 | 8258.89 | 65507.05 | – |
| | Backward [ms] | 3.17 | 28.46 | 170.88 | 1126.46 | 8374.37 | 129385.97 | – |
| | Total [ms] | 13.20 | 67.55 | 407.13 | 2755.28 | 16628.15 | 195462.18 | – |
| CVXPYLayers (SCS) | Accuracy | $1.69 \times 10^{-1}$ | $2.18 \times 10^{-1}$ | $2.97 \times 10^{-1}$ | Failed | Failed | Failed | Failed |
| | Forward [ms] | 94.70 | 510.04 | 1644.47 | – | – | – | – |
| | Backward [ms] | 36.27 | 167.21 | 404.12 | – | – | – | – |
| | Total [ms] | 129.26 | 683.34 | 2048.34 | – | – | – | – |

### E.1.2 Projection onto chain

For projection onto chains with links of length bounded by 1 in $\infty$-norm, as formulated in $P_2$, equation (9), the input point cloud $x_1, ..., x_m \in \mathbb{R}^d$ is set with $x_i \sim \mathcal{N}(0, 100 I_d)$, where the number of points is $m = 100$. By varying the dimension of the vector, $d$, we generated a 300 problem dataset with dimensions $n \in \{200, 500, 1000, 2000, 4000, 10000, 100000\}$. For $n \leq 4000$, each dimension contains 50 problems; for $n > 4000$ each dimension contains 25 problems. Gurobi is chosen as dQP's forward solver. Figure 8 and Table 6 demonstrate the solvers have performance similar to the projection onto the probability simplex shown in Figure 1, in terms of efficiency. Additionally, dQP successfully solves large-scale problems that other solvers fail to solve.

### E.1.3 Random sparse/dense problems

We generated two datasets of random QPs: sparse and dense.

For the random sparse dataset, $P = L^T L$, where $L$ is the standard Laplacian matrix of $k$-nearest graph ($k = 3$). Entries of $C$ and $A$ are filled by $\mathcal{N}(0, 1)$ random numbers with density of $5 \times 10^{-4}$; entirely zero rows are avoided. The vectors $d$ and $b$ are generated so that the constraints are feasible $d = C\mathbb{1} + \mathbb{1}$ and $b = A\mathbb{1}$. The dataset contains 625 problems with dimensions spanning $n \in \{100, 220, 450, 1000, 2100, 4600, 10000\}$, where $m = n$ and $p = n/2$. For $n \leq 4600$, each dimension contains 100 problems; for $n > 4600$ each dimension contains 25 problems. The KKT

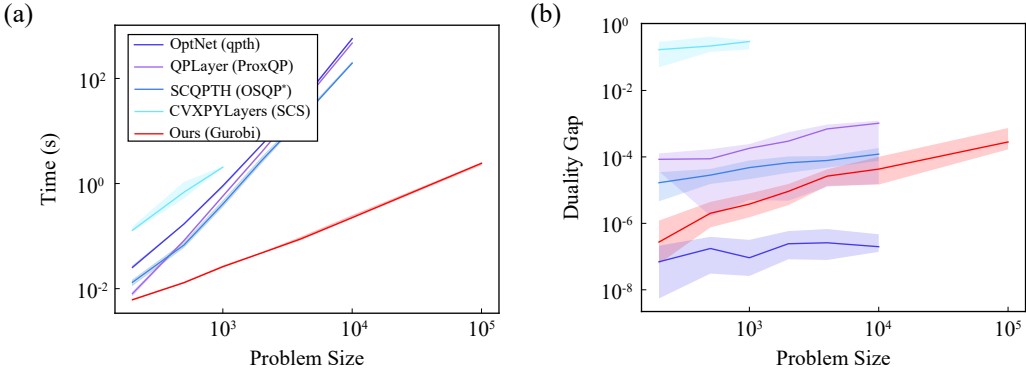

Figure 8: Time and accuracy performance for projection onto chains.

systems in these problems tend to be ill-conditioned. We use Gurobi as dQP's forward solver and employ a least-squares solver for the backward. In our experiments, shown in Table 7, OptNet and CVXPYLayers fail on all problems and SCQPTH is substantially slower, failing for $n \geq 4600$. dQP demonstrates superior accuracy and efficiency over QPLayer.

For the random dense dataset, $P = Q^T Q + 10^{-4} I$, where $Q \in \mathbb{R}^{n \times n}$ with $Q_{ij} \sim \mathcal{U}(0, 1)$, $C \in \mathbb{R}^{m \times n}$ with $C_{ij} \sim \mathcal{U}(0, 1)$, , and $A \in \mathbb{R}^{p \times n}$ with $A_{ij} \sim \mathcal{U}(0, 1)$. The constraint vectors are chosen in the same way as the sparse case. The dataset contains 450 problems with dimensions spanning $n \in \{10, 20, 50, 100, 220, 450, 1000, 2100, 4600\}$, $m = n$, and $p = n/2$. Each dimension contains 50 problems. We use DAQP and ProxQP as dQP's forward solvers. In the regime of these dense problems, qpth and ProxQP—the forward solvers used by OptNet and QPLayer, respectively—are highly competitive with other available solvers, both free and commercial. As such, we do not expect dQP to yield substantial gains. Indeed, Table 8 shows that our method is comparable to OptNet and QPLayer in both runtime and accuracy. For smaller dimensions ($n \leq 1000$), DAQP offers higher accuracy, while for larger problems, ProxQP proves more efficient.

Table 7: Time and accuracy performance statistics on random sparse problems.

| Solver | Metric | Problem Size | | | | | | |
|---|---|---|---|---|---|---|---|---|
| | | 100 | 220 | 450 | 1000 | 2100 | 4600 | 10000 |
| dQP (Gurobi) | **Accuracy** | $4.46 \times 10^{-8}$ | $9.23 \times 10^{-8}$ | $1.34 \times 10^{-7}$ | $6.89 \times 10^{-7}$ | $1.34 \times 10^{-6}$ | $3.16 \times 10^{-6}$ | $3.43 \times 10^{-6}$ |
| | Forward [ms] | 2.57 | **3.44** | **5.53** | **11.07** | **60.68** | **2446.70** | 143209.89 |
| | Backward [ms] | 1.79 | 2.86 | **4.73** | **9.70** | **24.03** | **309.10** | 9364.61 |
| | Total [ms] | 4.37 | **6.33** | **10.28** | **20.72** | **90.01** | **2760.07** | 151471.27 |
| OptNet (qpth) | | Failed | Failed | Failed | Failed | Failed | Failed | Failed |
| QPLayer (ProxQP) | Accuracy | $6.46 \times 10^{-6}$ | $1.25 \times 10^{-5}$ | $1.69 \times 10^{-5}$ | $3.04 \times 10^{-5}$ | $6.12 \times 10^{-5}$ | $1.77 \times 10^{-3}$ | $7.82 \times 10^{-5}$ |
| | Forward [ms] | **1.04** | 5.47 | 31.11 | 235.00 | 2268.24 | 23597.22 | 199009.91 |
| | Backward [ms] | **0.30** | **1.17** | 7.46 | 51.00 | 393.68 | 3538.53 | 38466.29 |
| | Total [ms] | **1.34** | 6.63 | 38.56 | 285.99 | 2658.82 | 27133.19 | 240084.62 |
| SCQPTH (OSQP*) | Accuracy | $2.21 \times 10^{-7}$ | $3.96 \times 10^{-7}$ | $1.64 \times 10^{-6}$ | $2.85 \times 10^{-5}$ | $1.32 \times 10^{-1}$ | Failed | Failed |
| | Forward [ms] | 4.43 | 8.36 | 27.64 | 4153.39 | 79627.33 | - | - |
| | Backward [ms] | 1.37 | 4.92 | 25.99 | 177.81 | 1588.04 | - | - |
| | Total [ms] | 5.82 | 13.38 | 53.84 | 4331.38 | 81211.59 | - | - |
| CVXPYLayers (SCS) | | Failed | Failed | Failed | Failed | Failed | Failed | Failed |

## E.2 Bi-Level Geometry Optimization

The geometry experiments were run on an Intel(R) Core(TM) i7-8850H CPU @ 2.60GHz with 6 cores.

We include a supplementary example in Figure 9 which illustrates the boundary conditions (inequality constraints in (10)) of [68] using blue arrows (corresponding to applying the Laplacian to the output vertices) and cones at points where the shape is locally non-convex. If the arrows fall outside the cones as shown in (b), then the map is non-invertible; if instead they lie strictly inside the cones then the map is invertible as shown in (c). The process is also flexible: by adding a regularizing term $\lambda \| \frac{M}{\|M\|_F} - \frac{M_c}{\|M_c\|_F} \|_\infty$ that measures the distance to the combinatorial Laplacian $M_c$ shown in (d),

Table 8: Time and accuracy performance statistics on random dense problems.

| Solver | Metric | Problem Size | | | | | |
|---|---|---|---|---|---|---|---|
| | | 20 | 100 | 450 | 1000 | 2100 | 4600 |
| dQP (daqp) | Accuracy | $\mathbf{1.59 \times 10^{-11}}$ | $\mathbf{1.20 \times 10^{-8}}$ | $2.35 \times 10^{-6}$ | $4.08 \times 10^{-5}$ | $5.26 \times 10^{-4}$ | Failed |
| | Forward [ms] | 0.20 | 1.31 | 131.35 | 1115.62 | 10065.77 | - |
| | Backward [ms] | **0.14** | 0.48 | 11.22 | 56.90 | 313.90 | - |
| | Total [ms] | 0.34 | 1.81 | 144.91 | 1174.47 | 10379.68 | - |
| dQP (proxqp) | Accuracy | $4.71 \times 10^{-6}$ | $6.42 \times 10^{-5}$ | $9.11 \times 10^{-4}$ | $7.26 \times 10^{-4}$ | $4.13 \times 10^{-4}$ | $4.25 \times 10^{-4}$ |
| | Forward [ms] | 0.29 | 2.54 | 61.12 | **379.74** | **2553.82** | **26408.12** |
| | Backward [ms] | 0.17 | 1.85 | 13.53 | 70.25 | 385.04 | 3369.77 |
| | Total [ms] | 0.46 | 4.32 | 73.68 | **455.22** | **2935.93** | **29771.33** |
| OptNet (qpth) | Accuracy | $6.89 \times 10^{-8}$ | $2.51 \times 10^{-8}$ | $\mathbf{3.80 \times 10^{-8}}$ | $\mathbf{3.51 \times 10^{-7}}$ | $\mathbf{2.80 \times 10^{-6}}$ | $\mathbf{3.34 \times 10^{-5}}$ |
| | Forward [ms] | 2.99 | 7.09 | 78.56 | 463.04 | 3176.59 | 29387.34 |
| | Backward [ms] | 0.23 | 0.45 | **5.55** | **29.30** | **185.80** | **1540.07** |
| | Total [ms] | 3.22 | 7.56 | 84.20 | 491.57 | 3362.25 | 30931.00 |
| QPLayer (ProxQP) | Accuracy | $3.08 \times 10^{-6}$ | $6.88 \times 10^{-5}$ | $3.98 \times 10^{-5}$ | $1.31 \times 10^{-4}$ | $1.35 \times 10^{-5}$ | $1.48 \times 10^{-4}$ |
| | Forward [ms] | **0.14** | **0.99** | **43.11** | 407.77 | 3973.89 | 43740.91 |
| | Backward [ms] | 0.15 | **0.34** | 9.67 | 74.24 | 601.25 | 5781.58 |
| | Total [ms] | **0.29** | **1.35** | **52.99** | 482.17 | 4575.13 | 49558.44 |
| SCQPTH (OSQP*) | Accuracy | $3.48 \times 10^{-5}$ | $4.62 \times 10^{-4}$ | $4.32 \times 10^{-5}$ | $6.54 \times 10^{-5}$ | $1.83 \times 10^{-4}$ | $2.26 \times 10^{-3}$ |
| | Forward [ms] | 10.01 | 26.72 | 120.12 | 664.74 | 6802.36 | 384565.40 |
| | Backward [ms] | 0.47 | 1.28 | 26.80 | 184.22 | 1733.72 | 15203.05 |
| | Total [ms] | 10.50 | 27.90 | 147.25 | 850.37 | 8550.04 | 399699.87 |
| CVXPYLayers (SCS) | Accuracy | $5.40 \times 10^{-4}$ | Failed | Failed | Failed | Failed | Failed |
| | Forward [ms] | 3.15 | – | – | – | – | – |
| | Backward [ms] | 1.08 | – | – | – | – | – |
| | Total [ms] | 4.22 | – | – | – | – | – |

we can enhance map quality in the sense that the triangles change their shape less with respect to the input mesh (a).

The upper-level loss for the bi-level cross experiment is shown in Figure 10(a) where the unregularized loss is driven to the desired tolerance, accompanied by sudden changes in the active set as the dual variables are driven to zero. We terminate the optimization at convergence, once all constraints are inactive to guarantee an injective map. For the regularized optimization (Figure 10(b)), we choose the regularization hyperparameter to be $\lambda = 10$ after sample testing. Despite the regularization, the dual loss can still be driven to 0 to reach an injective map.

To optimize over Laplacians $M$, we directly parameterize the space of Laplacians; we impose that the diagonals are the absolute row sums during optimization and that the off-diagonals are negative. We also constrain $M$ to have the same sparsity pattern as the combinatorial Laplacian. Lastly, since the Laplacian $M$ is the quadratic term in 1 the resulting QP is not strictly convex because the Laplacian does not have strictly positive eigenvalues. To address this, we perturb $M$ by a small scaling of the identity $10^{-4}I$.

Throughout the geometry experiments, we use the same solution tolerance $\epsilon_{\text{abs}} = 10^{-5}$ and active tolerance $\epsilon_J = 10^{-4}$ with the forward solver PIQP as determined by our tool for choosing the best solver described in Appendix D. For the upper-level optimization, we use the Adam optimizer with learning rate $10^{-2}$ [67]. We initialize the bi-level optimization with $M_c$.

We report only forward (QP) time for OptNet, QPLayer and SCQPTH in Figure 5(b) because OptNet and SCQPTH do not output, nor differentiate the duals, and while QPLayer does, it suffers poor scaling from dense operations. The backward timing that we report for dQP excludes any contribution coming from the set-up of the parameterized Laplacian and directly report the time to solve the reduced KKT and extract the gradients with respect to $M$.

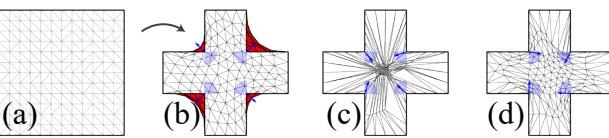

Figure 9: Computing an invertible map into a non-convex plus sign. A naive mapping from (a) the square into (b) the plus without cone constraints has inversions (red). Including cone constraints (blue) and performing the bi-level optimization leads to (c) an invertible map which can be regularized to (d) an enhanced quality map.

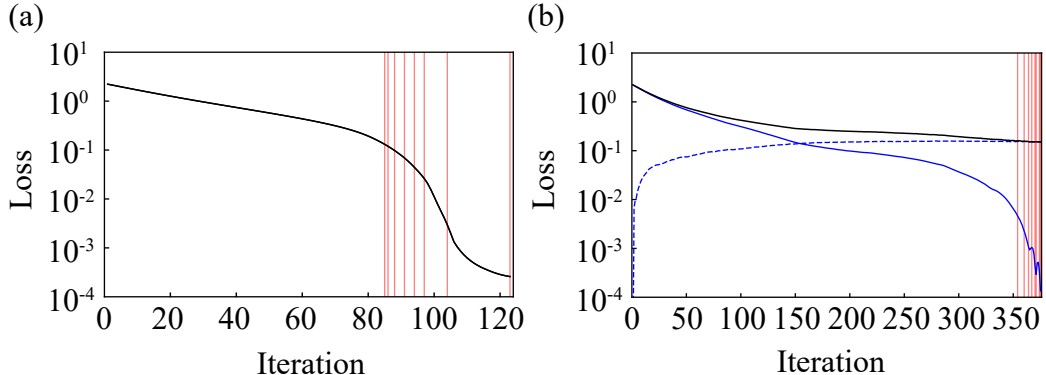

Figure 10: The evolution of the loss for the mappings of the square into the cross. Iterations for which the active set changes are denoted with vertical red lines. (a) Without regularization, the loss is driven monotonically to the tolerance. (b) With a competing regularizing loss term (dashed), convergence to the tolerance is slowed but not prevented.

The cross (Figure 9) and ant (Figure 5(a)) meshes and boundary constraints are obtained from the datasets in [41]. We create the synthetic mesh refinement example in Figure 5(b) by perturbing the corner of a square mesh.

