# OpenReview forum: "Differentiation Through Black-Box Quadratic Programming Solvers"
_NeurIPS.cc/2025/Conference — NeurIPS 2025 poster_

### Official Review · Reviewer_dAwU · 2025-06-02

**Clarity:** 3
**Significance:** 3
**Originality:** 3
**Rating:** 4
**Confidence:** 4

**Summary:**

This manuscript focuses on differentiation of quadratic programming problems. They note that there are many QP solvers and many previous attempts try to force the QP into some specific solver technique and differentiate that approach. But, this can make it so you can't always use the best QP solver. Thus instead of forcing everything to use a potentially not so great solver, the authors derive an adjoint rule so that a black-box QP solver can be used and a single linear solve (via LDT factorization) can reverse for the derivative. They then go on to show that if you use the right QP solver for each problem its faster than using a not so good one.

**Questions:**

1. You support 15 commercial solvers and 1 open source one with your new package: in Figure 4 when you say "ours", which QP solver is that using? It's ambiguous as to what exactly was used to solve there since that refers to your package and differentiation rule, but since there are 16 solvers it's not clear which one, or if that makes a difference.
2. Why not benchmark with the 16 different solvers together to better explain in which regime the given methods do better/worse?
3. Are there any domains where the other differentiation techniques do better? Obviously not in the large scaling case, but maybe there are cases that could be highlighted where not treating it as a black-box actually enables some optimization?
4. The method in the paper assumes zero residual of the forward solve and uses the solution value for the derivative. If there is an error in the forward solve, how does that error propagate into the derivative? Does it scale like the square root of the condition number of the Jacobian matrix being factorized? Does using the LDL / Cholesky could be improving the numerical accuracy by decreasing the growth here vs LU/QR?

**Ethical Concerns:**

["NO or VERY MINOR ethics concerns only"]

**Final Justification:**

I think it's pretty clear from the author responses that all of the good properties from the method are likely to come from just using a better QP solver and nothing really to do with their algorithm. Which is fine, but it should be better clarified in the paper. That makes it borderline, but since it's the right algorithm to do and QP solvers aren't doing this, it is at least acceptable.

**Limitations:**

Yes

**Quality:**

3

**Strengths And Weaknesses:**

The strength and the weakness is kind of the same thing here... it's pretty obviously the way to do this? I can easily see the mindset of the authors here. You scour the literature and notice that things like JaxOpt is differentiating the QP by transforming it into a convex optimization and using CVXPY, which yes you can always solve a QP by using a convex optimization solver because QPs are a subset but no one does that because specializing on the quadratic properties makes the solve much faster and more robust. And Alt-Diff's custom ADMM method is clearly way slower than actually good QP solvers like commercial solvers (e.g. Gurobi). So clearly the right thing to do is to just define the derivative of a QP solve and use a good solver and it will just be a whole lot faster than these bad QP solvers. So they did that and got the expected results. QED.

Is that novel enough? I don't really know. What's more surprising really is that the other authors were allowed to get away without this as the standard baseline test. The ADMM method and other papers, they should have been asked "test your method against defining the implicit differentiation rule over Gurobi and show the performance difference", and when it failed to be anywhere close to efficient that should've been the end of the story. In most other areas of differentiable programming, this has always been the baseline. You'd have to test differentiating through an ODE solver, nonlinear solver, etc. vs deploying a black-box rule that covers all solvers, and these black-box rules are implemented over lots of solvers in different domains already. I double checked the literature here and yes, it seems in QP solvers people just didn't do this, which is really odd, so the authors are doing the right thing that should've been done the first time like in all of the other domains. Is that novel enough for a NeurIPS publication? It is clearly the right thing to do and the best method, but most practitioners would have guessed that before reading it and its standard exercise to derive and define that adjoint rule, so there's nothing that is really shocking here other than the fact that this wasn't already done on QPs. Thus I am on the fence as to novelty, it's a strength that it's the clear best method to do but it's a weakness that it's not unobvious.

The fact that the adjoint is a linear solve is not surprising: the implicit differentiation theorem proves that all such cases of solvers which iterate to convergence has an adjoint rule that is given as a linear system. And the derivation of this linear system from the KKT equations is straightforward and standard. That's all run of the mill graduate-level homework these days. But, by subsetting it to the active set, noticing that gives a symmetric solve and thus can use an LDL factorization, that is big insight that makes this a lot more efficient over the naive application of the implicit differentiation rule. Thus I would point to this as the novelty of the paper, not what is mentioned in the title or abstract about supporting any black-box QP solver and that being the right way to do this, but this trick within the adjoint definition is a nice little trick.

---

> ### Author Rebuttal · Authors · 2025-07-29
>
> We thank the reviewer for their thorough summary and review. We address their questions below.
> > You support 15 commercial solvers and 1 open source one with your new package: in Figure 4 when you say "ours", which QP solver is that using? It's ambiguous as to what exactly was used to solve there since that refers to your package and differentiation rule, but since there are 16 solvers it's not clear which one, or if that makes a difference.
>
> A1: In Figure 4, “Ours (QPBenchmark)” means that for each problem we select the QP solver  that performed the best in the open-source benchmark, QPBenchmark. We noted this in  L277-279 and specified the solver counts in the caption.
>
> We use the open-source QPSolvers as a unified interface, allowing dQP to support all solvers it provides, currently including three commercial solvers (Gurobi, MOSEK, NPPro) and several open-source ones. QPSolvers is regularly updated with new solvers, which dQP inherits automatically.
>
> We will clarify these points in the text.
>
> > Why not benchmark with the 16 different solvers together to better explain in which regime the given methods do better/worse?
>
> A2: We found that including all ~2000 dQP variants with every supported solver in Figure 4 would have shifted the focus toward QP solver comparisons (covered extensively by QPBenchmark) and obscured the modularity benefits. We will revise the caption of Figure 4 and related text to better clarify.
>
> > Are there any domains where the other differentiation techniques do better? Obviously not in the large scaling case, but maybe there are cases that could be highlighted where not treating it as a black-box actually enables some optimization?
>
> A3: Yes, as we mention in L263-265. For example, OptNet’s reuses the internal factorizations in their custom GPU-batchable forward solver in their backward, which is not possible with black-box solvers. As a result, they can solve very small dense QPs in batches without a backward differentiation bottleneck, but cannot scale up to large, sparse problems.
>
> > The method in the paper assumes zero residual of the forward solve and uses the solution value for the derivative. If there is an error in the forward solve, how does that error propagate into the derivative?
>
> A4: The analysis of error propagation is complicated by the combinatorial quality of the active set that is crucial to dQP’s algorithm. If the problem is well-conditioned and the solvers used are sufficiently accurate so that the active set identified is locally stable, the error propagation to the derivatives boils down to the sensitivity of linear systems to perturbations, characterized by the condition number like the reviewer notes [24].
>
> [24] Blondel et. al., Efficient and Modular Implicit Differentiation, Theorem 1

---

> > ### Comment · Reviewer_dAwU · 2025-08-03
> >
> > "A2: We found that including all ~2000 dQP variants with every supported solver in Figure 4 would have shifted the focus toward QP solver comparisons (covered extensively by QPBenchmark) and obscured the modularity benefits. We will revise the caption of Figure 4 and related text to better clarify."
> >
> > I think that is a pretty mandatory part of the paper to understand the results. You say you use commercial solvers (Gurobi, MOSEK, NPPro), those are known to be a lot faster than the other QP solvers, is the benefit here only from those solvers? If you're simply benchmarking with the best QP solver in each case, is all of the benefit from picking a better QP solver? That's not necessarily a bad thing, but it's hidden in the benchmarks. Not including a table that answers the question makes me pessimistic have to assume that the author's approach is effectively no different and all of the performance difference comes from picking a commercial QP solver. If they want to convince me otherwise, they need to include a table that shows the other methods vs their method with the same QP solver as the other methods vs their method with the best QP solver. It doesn't need to be all 2000 variants, just two or three columns showing the other QP solvers to match what the others are doing in that respect. My guess is that the results which use the same QP solvers as the other methods are just average and the whole difference is just from a good QP solver.
> >
> > I think there's nothing wrong with that being the result, I think the paper is still fine and publishable if that's the case. I just don't see why this needs to be hidden. Make it clear whether or not that's the case by sharing the benchmark that distinguishes how much of the difference is the QP solver and how much of the difference is other parts of the method. The purpose of benchmarks is not to be the best, it's to learn and share what matters for performance, and the benchmarks here clearly do not achieve the goal of clarity because there are multiple factors at play and the result cannot be attributed to any single piece in the way it's shared.
> >
> > "A3: Yes, as we mention in L263-265. For example, OptNet’s reuses the internal factorizations in their custom GPU-batchable forward solver in their backward, which is not possible with black-box solvers. As a result, they can solve very small dense QPs in batches without a backward differentiation bottleneck, but cannot scale up to large, sparse problems." Thanks for clarifying, indeed it matches expectations I just missed those lines.
> >
> > > A4: The analysis of error propagation is complicated by the combinatorial quality of the active set that is crucial to dQP’s algorithm. If the problem is well-conditioned and the solvers used are sufficiently accurate so that the active set identified is locally stable, the error propagation to the derivatives boils down to the sensitivity of linear systems to perturbations, characterized by the condition number like the reviewer notes [24].
> >
> > It would be good to add a note in the paper about the numerical properties. While that's usually omitted from ML work so not required, it would make it strong to make this aspect clear to readers.

---

> > > ### Author Response · Authors · 2025-08-05
> > >
> > > We thank the reviewer for their comments and we hope that the points below will help clarify some issues raised:
> > > * Runtimes depend on both the QP solver (forward) and differentiation (backward). We report their statistics separately in Tables 1 and 5–8.
> > > * Other comparable differentiation methods, e.g. OptNet, are built on and work with particular QP solvers. Therefore, their backward time can only be measured with these solvers.
> > > * dQP enables differentiating arbitrary QP solvers. While fast QP solvers are key to dQP’s performance, we also demonstrate performance gains in the backward in certain scenarios (e.g., Table 1, Column 9 “Avg Bwd”).
> > > * Open-source solvers can outperform commercial solvers (e.g., in ~92% of the problems in Table 1 and Figure 4), while in other cases, such as in Figure 1, commercial solvers perform better. dQP provides an interface to both.
> > >
> > > We are happy to include the full experimental data, including separate forward and backwards times for each instance, in the project code repository for complete transparency.

---

### Official Review · Reviewer_EkQp · 2025-06-20

**Clarity:** 3
**Significance:** 2
**Originality:** 2
**Rating:** 4
**Confidence:** 3

**Summary:**

This work studies how to compute the gradients of QP solutions with respect to QP problem parameters, making QP solve a differentiable layer that can be potentially integrated into a larger computational graph. The problem is a classic, extensively studied topic in sensitivity analysis. The proposed method obtains QP solutions from an external QP solver and analyzes the KKT conditions that connect the QP solution to the problem parameters. One of the key ingredients in the technical method is the reduced dimensionality analyzed from the active set of QP constraints. The paper provides an open-source implementation and evaluates its performance against several differentiable QP implementations in PyTorch.

**Questions:**

Perhaps my only major concern with this paper is the novelty of the technical method. I will first list my understanding of the contribution #1 (lines 57-58) in the paper, which seems to focus on two key claims:
1. The paper derives the gradients of a QP solver from its primal solution without dependency on the exact QP solver itself. This leads to the solver-agnostic feature of dQP, which can be used with any existing QP solvers.
2. The paper uses the set of active constraints at the solution to simplify the dimension of the system solved for gradient computation, which seems to be rooted in the standard implicit function theorems/adjoint methods applied to the KKT system of the QP problem.

For #1: Decoupling backward differentiation from its forward solver is not a novel thing in sensitivity analysis, e.g., previous works have explained it when differentiating Navier-Stokes flows [Stuck 2012]. This fact is a natural result of the implicit function theorem and the classic adjoint method (Bradley et al. [2004]).

For #2: Pruning inactive constraints is a standard practice in differentiable programming when a solver contains constraints. A concrete example is differentiable simulation with a Columb contact and friction model [Li et al. 2022], where inactive constraints from the complementarity conditions in the Columb model are pruned in gradient computation.

Therefore, while I agree that dQP may be novel for differentiating QP problems, I feel it is also an expected and predictable result from existing theories, as similar patterns from common practice in sensitivity analysis/differentiable programming can be found in existing works that differentiate other numerical solvers. This shakes my confidence in the significance of the proposed method. My questions to the authors are:
1. Could you summarize what new knowledge this paper brings to differentiable programming/sensitivity analysis?
2. Is there anything QP-specific that can be highlighted or exploited in developing your technical method?

References
- Andrew Bradley, PDE-constrained optimization and the adjoint method, 2004.
- Arthur Stuck, Adjoint Navier-Stokes methods for hydrodynamic shape optimisation, 2012.
- Yifei Li, et al. DiffCloth: Differentiable Cloth Simulation with Dry Frictional Contact, 2022.

**Ethical Concerns:**

["NO or VERY MINOR ethics concerns only"]

**Final Justification:**

The rebuttal's answers to my questions are more or less expected. Technical innovation is not the strength of this work. Instead, it finds and fills a gap in differentiating QP with a well-integrated system. This was roughly what I thought when I was writing my original review, so the rebuttal does not significantly affect my rating either. I am comfortable with maintaining my borderline accept rating.

**Limitations:**

I appreciate that the paper is upfront about the lack of GPU support. I am actually a bit surprised to learn that “state-of-the-art sparse and scalable QP solvers with GPU support are still lacking.” I thought QP algorithms are largely based on solving linear systems of equations, and large-scale linear solvers have GPU support these days. I am taking this sentence at face value and won't let this claim affect my evaluation of this work negatively, but I am curious to learn more about this claim if the authors are willing to elaborate on it.

**Paper Formatting Concerns:**

None.

**Quality:**

3

**Strengths And Weaknesses:**

Overall, I like the direction this paper explores, and I want to commend the open-source implementation.

**Strengths**

1. I appreciate that the paper offers an open-source implementation of the proposed algorithm. I think the engineering effort and the modular design look valuable to me.

2. The experiments on many baselines and benchmark problems seem comprehensive. I think these experiments have covered some widely accepted and fairly strong numerical solvers, e.g., Gurobi, Pardiso, and cvx.

**Weaknesses**

The novelty and significance of the contribution “QPs can be explicitly differentiated using only the primal solution via a reduced dimension locally equivalent linear system determined by the active set” is unclear to me. I will leave my specific questions in the “Questions” section below.

---

> ### Author Rebuttal · Authors · 2025-07-29
>
> We thank the reviewer for their thorough summary and review. We address their questions below.
>
> > Decoupling backward differentiation from its forward solver is not a novel thing in sensitivity analysis … Could you summarize what new knowledge this paper brings to differentiable programming/sensitivity analysis? Is there anything QP-specific that can be highlighted or exploited in developing your technical method?
>
> A1: We agree with the reviewer that we did not introduce the general idea of separating backward differentiation from the forward solver. However, in the context of QPs, prior methods often tightly integrate differentiation with specific solvers, e.g., to enable information sharing between the forward and backward. As we acknowledge in Section 3 and the related work section, components of our approach (e.g., pruning inactive constraints) have appeared before, but no prior work has synthesized these ideas into a fully modular differentiable QP layer. Our contribution lies in combining these elements to provide a clear, flexible framework, and in making the key observation that, for QPs, a single symmetric solve over the active set can yield both dual variables if needed and derivatives. Practically, we also demonstrate that for certain problems, especially large-scale, sparse ones, the benefits of using efficient forward solvers far outweigh any gains from solver-specific integration, supporting the case for modularity over tight coupling.
>
> > I appreciate that the paper is upfront about the lack of GPU support. I am actually a bit surprised to learn that “state-of-the-art sparse and scalable QP solvers with GPU support are still lacking.” I thought QP algorithms are largely based on solving linear systems of equations, and large-scale linear solvers have GPU support these days. I am taking this sentence at face value and won't let this claim affect my evaluation of this work negatively, but I am curious to learn more about this claim if the authors are willing to elaborate on it.
>
>
> A2: We are happy to elaborate, though we are not experts on this topic. Until recently, Gurobi maintained that the potential benefits of GPU-based solvers were minimal, guiding our own view. A key challenge is that large-scale problems of interest are typically sparse, and efficiently exploiting sparsity on GPUs remains a difficult and developing area. Recently, Gurobi and NVIDIA began collaborating on a GPU-based solver, “cuOpt.” Gurobi documents the latest and previous updates on their efforts to use GPUs in optimization in their forum post, “Does Gurobi support GPUs?”

---

> > ### Comment · Reviewer_EkQp · 2025-08-03
> > **Rebuttal response**
> >
> > I think the rebuttal confirms my conjecture of the paper's technical novelty: it mainly focuses on integrating existing techniques known in other fields. I strongly align with the opinions expressed in the Strengths and Weaknesses in Reviewer dAwU's review: it is a standard and obvious idea in differentiable programming for many other problem domains, and it is odd that the QP community hasn't tried what this submission suggested before, according to the paper's literature review.
> >
> > While the rebuttal shakes my confidence in the paper's technical novelty, there is still sufficient merit in providing a functional and well-integrated solution, which I do appreciate and would like to encourage. Based on all these factors, I will maintain my borderline acceptance rating for now, but I don't feel strongly enough to argue for accepting this paper.

---

### Official Review · Reviewer_gwAp · 2025-06-25

**Clarity:** 4
**Significance:** 2
**Originality:** 2
**Rating:** 5
**Confidence:** 5

**Summary:**

The paper investigates how to compute gradients through black-box Quadratic Programming (QP) solvers to efficiently incorporate QP layers into neural networks or bilevel optimization tasks. It introduces dQP, a modular framework that enables automatic differentiation of any QP solver, transforming it into a differentiable layer. The core theoretical insight is based on a key observation: given a known active constraint set, the gradient of the QP solution can be explicitly expressed as a simplified linear system, significantly reducing computational complexity.

**Questions:**

* How is the duality gap defined in the context of dQP? Why is it used as a convergence metric? While the paper uses the duality gap to report solution quality, a precise formal definition (in this context) and explanation of why it reliably measures convergence across solver backends would be helpful. Does it account for both primal and dual feasibility in the reduced KKT system?

* While dQP is clearly more flexible, I am still unclear on the primary motivation from a user perspective. Why would users choose dQP over existing approaches like OptNet, which already perform quite well? In other words, what is the key intuition or motivation for dQP?

**Ethical Concerns:**

["NO or VERY MINOR ethics concerns only"]

**Final Justification:**

The author's response well addressed my concern, so I will maintain my original score and vote for acceptance.

**Quality:**

3

**Strengths And Weaknesses:**

# Strengths
* The paper provides an extensive and rigorous investigation of differentiable quadratic programming (QP), situating its contributions within a rich body of related work. The authors evaluate their method across a wide variety of benchmarks, including dense and large-scale sparse QPs, and demonstrate dQP’s superior scalability and robustness.
* A major strength is the explicit formulation of the differentiation strategy based on the active set. The authors clearly connect sensitivity analysis, the KKT conditions, and the derivation of a reduced KKT system. This makes the framework both conceptually clean and practically efficient.
*  The proposed dQP framework is notable for supporting a wide range of existing QP solvers via a modular design. It decouples differentiation from the solver itself, enabling plug-and-play usage with over 15 solvers, including commercial and open-source ones, which significantly broadens the applicability of differentiable optimization.


# Weakness
* The paper reports timing comparisons in milliseconds. However, the differences—especially in the context of total runtime—may amount to only a few seconds even for large-scale problems. It would help if the authors could clarify the scenarios or applications (e.g., bi-level learning or large-scale simulation) where these runtime savings are practically consequential.

* The paper demonstrates excellent performance on a wide benchmark, but there is little discussion of potential failure modes (e.g., sensitivity to incorrect active set identification, degeneracy, or ill-conditioning near singularities). While Appendix D mentions heuristics, it would strengthen the paper to include a quantitative evaluation of robustness in such edge cases.

Overall, I think this is a well-written paper currently.

---

> ### Author Rebuttal · Authors · 2025-07-29
>
> We thank the reviewer for their thorough summary and review. We address their questions below.
>
> > How is the duality gap defined in the context of dQP? Why is it used as a convergence metric? While the paper uses the duality gap to report solution quality, a precise formal definition (in this context) and explanation of why it reliably measures convergence across solver backends would be helpful. Does it account for both primal and dual feasibility in the reduced KKT system?
>
> A1: We define the duality gap and residuals for QP in Appendix E.1 Lines 742-748. We use the duality gap measure for accuracy as it readily provides an upper bound on the objective error (with respect to the true unknown optimal objective value) [27]. The duality gap is routinely used in benchmarks (e.g., QPBenchmark) and in the stopping criteria of many solvers. We separately measure primal and dual feasibility, and while they are not reported, they are used along with the duality gap, as described in Appendix E.1 Lines 751-756,  to determine “solver success.” We will clarify these points in the main text.
>
> [27] Boyd and Vandenberghe, Convex Optimization, Pages 241-242
>
> > While dQP is clearly more flexible, I am still unclear on the primary motivation from a user perspective. Why would users choose dQP over existing approaches like OptNet, which already perform quite well? In other words, what is the key intuition or motivation for dQP?
>
> A2: Users should choose dQP for large-scale, sparse problems, where its ability to leverage a wide range of commercial solvers (as noted by the reviewer) can yield orders-of-magnitude speedups over other differentiable QP methods (Figure 1). The advantage of this flexibility is also evident in the MM experiment (Figure 4). While OptNet performs well on some problems, it fails on 70% of the MM instances (Figure 4), underperforms on the rest (Table 1), and scales poorly for sparse problems (Figures 1, 5; Tables 2-3, 5-7). Other differentiable solvers face similar limitations.
>
> This also connects to the reviewer’s point about runtime differences: though small in isolation, these differences accumulate and become crucial (or even prohibitive) in bilevel optimization and learning applications. We thank the reviewer for pointing this out and will clarify it in the revision.

---

> > ### Comment · Reviewer_gwAp · 2025-08-04
> >
> > Thanks for the response. I believe the current version of this paper is indeed good enough to be accepted and I will maintain my original score.

---

### Official Review · Reviewer_13br · 2025-07-02

**Clarity:** 3
**Significance:** 3
**Originality:** 2
**Rating:** 5
**Confidence:** 3

**Summary:**

This paper provides a modular approach, called dQP, for differentiable optimization through QP. Unlike existing approaches, dQP can be equipped with any solver. Once the solution is identified, the QP system is reduced to equality constraint problems by restricting to the active constraints. Therefore, the differentiation operation is reduced to solving a simple linear system. Experiments show dQP is very promising, particularly for large-scale sparse systems.

**Questions:**

Can your framework be extended to handle linear programs or, more generally, convex programs in which the quadratic objective is replaced by a linear or other convex function? If not, could you clarify the specific technical limitations that prevent such an extension?

**Ethical Concerns:**

["NO or VERY MINOR ethics concerns only"]

**Final Justification:**

The rebuttal have addressed the proposed issue and i am generally ok. I will keep my score

**Limitations:**

Yes, the limitation has been properly addressed.

**Quality:**

3

**Strengths And Weaknesses:**

# Strengths
The manuscript is well structured and easy to follow, with each theoretical idea accompanied by intuitive explanations and supporting figures.  The experiments convincingly demonstrate both performance and scalability; in particular, dQP is the only method in the comparison that successfully solves very large‐scale QPs while maintaining competitive accuracy.


# Weaknesses

A potential limitation is the migration to GPU implementation. The current implementation is CPU-only. Porting the method to GPUs will almost certainly require iterative solvers for the reduced KKT system, as I highly doubt the speedup of direct linsysm solver in GPU; yet such iterative solvers often trade speed for numerical precision. Recent first-order, GPU-friendly QP solvers (e.g., PDQP (https://github.com/jinwen-yang/PDQP.jl).) suggest one possible path forward, but PDQP is written in Julia and cannot be plugged into PyTorch without significant engineering. A native GPU backend may therefore need to be developed from scratch.

Another remaining issue is how to handle degeneracy. The paper introduces an active-set refinement heuristic that appears effective in practice, but a deeper analysis of how solver accuracy influences active-set identification—especially in degenerate or near-degenerate regimes—would strengthen the contribution. While full theoretical treatment may be beyond the current scope, even an empirical study of pathological cases would add confidence in the method’s robustness.

---

> ### Author Rebuttal · Authors · 2025-07-29
>
> We thank the reviewer for their thorough summary, review, and questions.
>
> We agree that integration with GPU-based sparse iterative solvers for QP and linear systems will be highly desirable once such tools become widely available, as we noted in our conclusion. In the meantime, our work contributes a scalable CPU-based solution that enables the solution of large-scale problems beyond the reach of existing approaches.
>
> Some problems inherently exhibit sensitivity or ill-conditioning in the active set, leading to near-degeneracy in the KKT conditions. Nevertheless, as noted in lines 243–245, we used a fixed threshold to determine the active set across all our experiments. Our implementation provides supplementary tools to help users select solvers, tune parameters, and optionally apply an active-set refinement heuristic (Appendix D). The latter was not used in our experiments.
>
> > Can your framework be extended to handle linear programs or, more generally, convex programs in which the quadratic objective is replaced by a linear or other convex function?
>
> A1: Linear programs are challenging because their solutions are generically at extremal points, leading to complexities in differentiation that we, and most other differentiable QP solvers, do not directly address. We are currently working on extensions of the framework to more general classes of convex optimization, like described by the reviewer.

---

> > ### Comment · Reviewer_13br · 2025-08-04
> >
> > Thank you for the reply. My feedback is intended to offer suggestions rather than criticism. I will keep my score.

---

### Decision · Program_Chairs · 2025-09-17

**Decision:**

Accept (poster)

**Comment:**

This paper introduces dQP, a modular and solver-agnostic framework for plug-and-play differentiation of virtually any QP solver. The authors' open-source implementation integrates more than 15 state-of-the-art solvers. Extensive experiments comparing against other strong differentiable solvers demonstrate the accuracy and robustness of dQP on small- and large-scale settings, and in both sparse and dense regimes.

In some sense, the submission does not present anything conceptually new. The principles of designing QP layers via implicit differentiation through the KKT conditions, of active set differentiation, of using Jacobian-vector products, and of decoupling the forward pass from the backward pass are not new. (While the authors call their differentiation technique "explicit differentiation," this seems to be at odds with how the differentiable layers literature uses this term - this is still differentiation through the KKT conditions at the solution point, albeit using an equivalent problem, rather than differentiating through the unrolled iterations of the QP solver.) In addition, the solver presented is CPU-only and is not evaluated/usable in batched settings.

Despite this, the reviewers felt the paper makes a strong contribution through its software package. In particular, it is beneficial to allow the use of many state-of-the-art solvers, allowing the right solver for the problem to be chosen and for in-built tricks such as warm-starting to be leveraged. The package is indeed benchmarked thoroughly in non-batched settings, demonstrating good performance. The active-set refinement strategy is important in ill-conditioned settings, and other differentiable solvers do not necessarily provide this. There is support for sparse solvers, allowing for associated speedups, whereas the GPU-based solvers in the literature only support dense representations.

Overall, the strengths of the practical contribution (software package and extensive benchmarking) seem to outweigh the limited conceptual novelty, providing a case for acceptance. However, the authors should consider significantly updating the narrative of the paper to make the true contributions clearer (notably, to indicate that the main contribution and source of benefit is the software package that enables better/diverse solvers to be used in the forward pass, rather than anything about the backward pass algorithm).